# The receptor PTPRU is a redox sensitive pseudophosphatase

Iain M. Hay [1,2], Gareth W. Fearnley[1,2], Pablo Rios[3], Maja Köhn [3], Hayley J. Sharpe [1,2✉] & Janet E. Deane [1✉]

The receptor-linked protein tyrosine phosphatases (RPTPs) are key regulators of cell-cell communication through the control of cellular phosphotyrosine levels. Most human RPTPs possess an extracellular receptor domain and tandem intracellular phosphatase domains: comprising an active membrane proximal (D1) domain and an inactive distal (D2) pseudo-phosphatase domain. Here we demonstrate that PTPRU is unique amongst the RPTPs in possessing two pseudophosphatase domains. The PTPRU-D1 displays no detectable catalytic activity against a range of phosphorylated substrates and we show that this is due to multiple structural rearrangements that destabilise the active site pocket and block the catalytic cysteine. Upon oxidation, this cysteine forms an intramolecular disulphide bond with a vicinal "backdoor" cysteine, a process thought to reversibly inactivate related phosphatases. Importantly, despite the absence of catalytic activity, PTPRU binds substrates of related phosphatases strongly suggesting that this pseudophosphatase functions in tyrosine phosphorylation by competing with active phosphatases for the binding of substrates.

[1] Cambridge Institute for Medical Research, Hills Road, Cambridge CB2 0XY, UK. [2] Signalling Programme, Babraham Institute, Cambridge CB22 3AT, UK. [3] Signalling Research Centres BIOSS and CIBSS, and Faculty of Biology, University of Freiburg, Schänzlestr. 18, Freiburg D-79104, Germany. ✉email: Hayley.sharpe@babraham.ac.uk; Jed55@cam.ac.uk

The human classical protein tyrosine phosphatases (PTPs) are key signalling regulators that work with kinases to fine-tune cellular levels of phosphotyrosine (pTyr), impacting on multiple cellular pathways including metabolism, differentiation, and adhesion[1–3]. PTPs do not simply function as negative regulators of tyrosine kinases to reverse protein phosphorylation, instead it is becoming clear that PTPs work in synergy with kinases to regulate complex cell signalling pathways and are important therapeutic targets in diseases such as cancer and diabetes[4,5]. The 37 classical PTPs exhibit diverse domain architectures and subcellular localizations, but all share a conserved core catalytic $C-X_5-R$ motif, known as the PTP loop, which includes the essential cysteine that catalyses the nucleophilic attack on the substrate phosphate group[6,7]. Less well conserved are the three additional motifs that form the PTP active site[7]. The WPD loop contains an aspartate residue that acts as a general acid/base during different steps of the catalytic cycle and assists in substrate binding. The pTyr recognition loop, typically containing the amino acid sequence KNRY, is so called because it forms the deep pocket that imparts selectivity for pTyr over smaller phosphorylated amino acids, such as serine and threonine. In addition to defining the shape of the binding pocket, the tyrosine in the pTyr recognition loop plays a crucial role in substrate orientation as its sidechain packs against the substrate pTyr phenyl ring. Finally, the Q loop positions and activates a water molecule for the hydrolysis of the phosphocysteine intermediate complex.

Despite the importance of catalysis for the function of many PTPs, there are numerous reports of non-catalytic functions[8–10]. Moreover, 5 of the 37 classical PTPs have been reported to be catalytically inactive against generic phospho-substrates, such as pNPP, DiFMUP and phosphopeptides. These include the non-receptor PTPs: PTPN23 (HD-PTP)[11,12], PTPN14 (PTPD2) and PTPN21 (PTPD1)[11,13] and the receptor PTPs: PTPRN (PTPIA2) and PTPRN2 (PTPIA2β)[14]. These PTPs contain altered sequences in their catalytic motifs and substrate binding loops that are predicted to impair catalytic activity, defining them as putative pseudoenzymes[15]. For example, PTPN23 (HDPTP) has an incomplete Q loop and serine substitution within the PTP loop, PTPN21 (PTPD1) possesses an altered WPD motif and PTPN14 (PTPD2) has a variant pTyr recognition loop[11,13]. However, it is noteworthy that activity against specific protein substrates has been reported for some of these PTPs, raising the possibility of non-canonical activation mechanisms[16,17]. Beyond the non-receptor PTPs, 12 of the 21 cell surface receptor PTPs possess highly conserved membrane distal pseudophosphatase D2 domains, which have been implicated in substrate recognition, redox sensing and enzyme regulation[3,18,19]. Interestingly, changes in the catalytic motifs can also alter substrate specificity. For example, a glutamate substitution in the WPD motif of the single PTP domain of PTPRQ (PTPS31) determines its selectivity for phosphoinositides over pTyr[20]. These examples illustrate the importance of combining functional and structural studies to characterise the catalytic properties of putative pseudophosphatases.

PTPRU is a member of the R2B receptor family, which includes PTPRK, PTPRM and PTPRT, characterised by large extracellular domains that mediate homophilic interactions and tandem intracellular PTP domains (Fig. 1a)[21]. PTPRU is expressed during development[22,23], and is reported to function during zebrafish gastrulation[24] and chick midbrain development[25]. Interestingly, while PTPRK and PTPRT are reportedly tumour suppressors, PTPRU has been proposed to play an oncogenic role in gastric cancer and glioma cells[26,27]. Opposing reports suggest that both PTPRU overexpression and knockdown can lead to β-catenin dephosphorylation[26,28,29]. Recently, the PTPRK substrate Afadin[3] was identified as a PTPRU interactor[30], implicating it as a cell adhesion regulator. The membrane

proximal (D1) domains of PTPRK, PTPRM, and PTPRT are active tyrosine phosphatases[11], however, the catalytic activity of PTPRU has not been determined. PTPRU possesses evolutionarily conserved sequence changes to key catalytic motifs including non-canonical WPD (WPE) and pTyr recognition loop sequences (GSRQ rather than KNRY), as well as a unique threonine in the PTP loop (Fig. 1b and Supplementary Figs. 1 and 2). To better understand the function of PTPRU we set out to determine whether it is an active phosphatase.

Here, we demonstrate through biochemical and structural studies that PTPRU is unique amongst the receptor-linked protein tyrosine phosphatases (RPTPs) in possessing two pseudophosphatase domains. The crystal structure of the PTPRU D1 domain reveals substantial structural rearrangements to key catalytic loops such that the shape of the pTyr binding pocket is lost and the active site cysteine is occluded. Despite lacking catalytic activity, PTPRU can recruit substrates of catalytically active paralogs supporting a model where PTPRU functions as a scaffold to compete for binding to protein substrates. Thus, the levels of the different R2B RPTPs expressed at the plasma membrane will determine the local pTyr level of a subset of cell junction regulators.

## Results

**PTPRU is catalytically inactive.** To determine the consequence of sequence variations on phosphatase activity, we expressed and purified the recombinant PTPRU D1 domain in *E. coli* for use in in vitro phosphatase assays. As a positive control we used the D1 domain of its closest paralog, PTPRK (Supplementary Table 1). The generic substrate 4-nitrophenyl phosphate (pNPP) was used in initial activity assays. The $K_m$ and $k_{cat}$ for PTPRK-D1-mediated pNPP hydrolysis were $16.16 \pm 1.32$ mM and $4.50 \pm 0.17$ s$^{-1}$, respectively, similar to previously determined kinetic parameters for the prototypic phosphatase PTP1B using this substrate (Fig. 1c)[31]. Strikingly, we were unable to detect activity for PTPRU-D1 against pNPP, even at high enzyme concentrations (up to 25 μM) or when using an extended assay duration (Fig. 1d), whereas PTPRK-D1 activity was readily detectable in the low nanomolar range (Fig. 1d). To investigate whether a cellular cofactor might be necessary for PTPRU activity, we performed dephosphorylation assays using quenched pervanadate-treated cell lysates (see "Methods"). These lysates are enriched in tyrosine phosphorylated proteins that serve as substrates for recombinant PTP domains. Again, whilst incubation with PTPRK-D1 for 16 h at 4 °C resulted in visible dephosphorylation of total cellular pTyr, PTPRU-D1 showed no activity and is comparable to the inactive PTPRK D2 domain (Fig. 1e). One possible explanation for the inactivity of PTPRU could be a requirement for its D2 domain. Previous studies on other receptors have shown that interactions between the two domains can impact D1 activity[3,32]. Unfortunately, we were unable to purify the full PTPRU intracellular domain (ICD) from bacteria. Therefore, we generated N-terminal flag-tagged PTPRK and PTPRU ICD mammalian expression constructs, encompassing the juxtamembrane, D1 and D2 domains (Fig. 1f). ICDs were immunoprecipitated (IP) from transfected HEK-293T lysates and subjected to pNPP assays (Fig. 1g and Supplementary Fig. 3a). Whilst the PTPRK-ICD, but not an inactivating cysteine mutant (C1089S), was able to hydrolyse pNPP, PTPRU and the corresponding cysteine mutant showed no activity above mock IPs after 2 h (Fig. 1h and Supplementary Fig. 3b). Taken together, these data suggest that unlike PTPRK, the D1 domain of PTPRU has no intrinsic PTP activity.

Due to the highly divergent nature of the PTPRU pTyr recognition loop, we investigated whether PTPRU may have

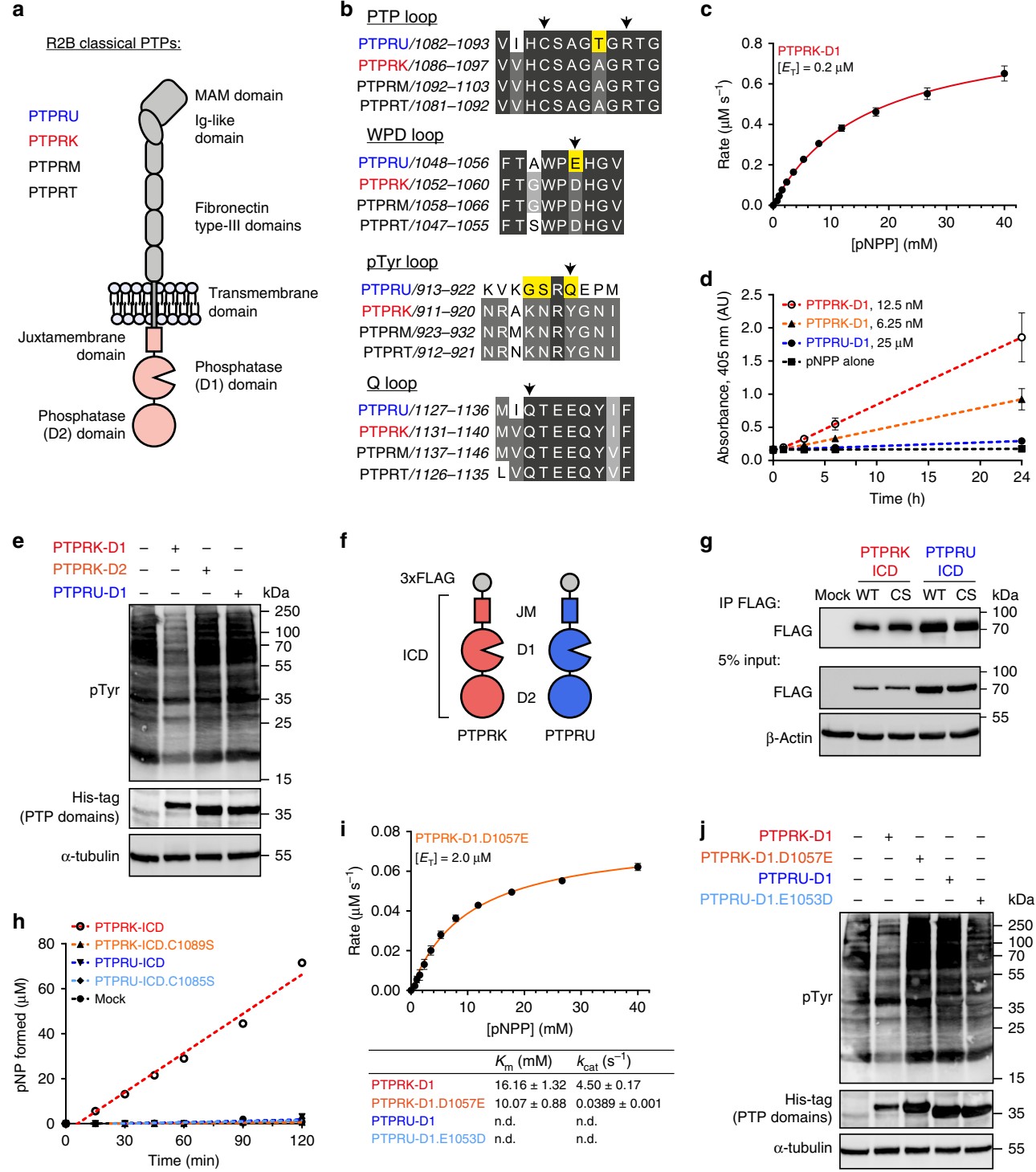

altered substrate specificity. PTPRU displayed no phosphatase activity against phosphoserine (pSer) or phosphothreonine (pThr; Supplementary Fig. 4a). Notably, the PTPRQ D1 domain has the same non-canonical aspartate to glutamate substitution in the WPD-loop observed in PTPRU (Supplementary Fig. 1), and has been previously reported to be a phosphoinositide phosphatase[20]. However, PTPRU-D1 exhibited no measurable dephosphorylation of either phosphatidylinositol (PI)-4-phosphate or PI-4,5-bisphosphate (Supplementary Fig. 4b). While our investigation of protein phosphorylation was primarily focused on residues modified with an O-linked, phosphoester bond (pTyr, pSer, pThr), a recent mass spectrometry study has reported residues modified with an N-linked, phosphoramidate bond (pHis, pLys, pArg, pAsp) as having greater abundance in the cell than pTyr[33]. Indeed, human histidine phosphatases have been identified previously[34–36]. To test the ability of PTPRU to catalyse hydrolysis of phosphoramidate bonds we used the generic histidine phosphatase substrate imidodiphosphate (PNP; Supplementary Fig. 4c)[37] but were again unable to detect any activity (Supplementary Fig. 4d). Together these data indicate that PTPRU does not catalyse the hydrolysis of phosphoester or phosphoramidate-based substrates.

In the absence of an identifiable substrate, we sought to determine the molecular mechanism of PTPRU-D1 inactivity.

**Fig. 1 The PTPRU D1 domain does not dephosphorylate pTyr. a** Schematic diagram of the R2B family RPTP domain structure. **b** Multiple sequence alignment of the 4 key PTP motifs across R2B family RPTPs, coloured by percentage identity (light-dark grey). Key variable residues in PTPRU are highlighted in yellow and essential PTP catalytic residues are marked by arrowheads. **c** Michaelis-Menten plot of initial rate vs. substrate (pNPP) concentration using 0.2 μM PTPRK-D1. Error bars represent ± SEM of $n = 3$ independent experiments. **d** Extended time course of pNPP dephosphorylation, monitored by absorbance at 405 nm, using low concentration (6.25 nM, orange, and 12.5 nM, red) PTPRK-D1 and high concentration (25 μM, blue) PTPRU-D1 recombinant proteins. pNPP substrate alone (black) is included as a control. **e** Immunoblot analysis of pervanadate-treated MCF10A lysates incubated with 0.2 μM PTPRK-D1, PTPRK-D2 or PTPRU-D1 recombinant PTP domains for 16 h at 4 °C. **f** Schematic of N-terminal FLAG-tagged PTP intracellular domain (ICD) constructs used in this study encompassing the juxtamembrane (JM) and D1 and D2 domains. **g** Immunoblot analysis of FLAG immunoprecipitations (IP) from HEK-293T cells transiently transfected with PTPRK and PTPRU WT and CS inactivating mutant (PTPRK; C1089S, PTPRU; C1085S) ICDs. **h** Time course of pNPP dephosphorylation following FLAG IP of ICDs as described in (**g**), showing PTPRK-ICD (red), PTPRK-ICD.C1089S (orange), PTPRU-ICD (blue), PTPRU-ICD.C1085S (light blue) and mock (black). **i** Michaelis-Menten plot of initial rate vs. substrate (pNPP) concentration using 2 μM PTPRK-D1.D1057E. Error bars represent ± SEM of $n = 3$ independent experiments. Kinetic parameters for PTPRU-D1 and PTPRU-D1.E1053D could not be determined (n.d.). Table shows kinetic parameters $K_m$ and $k_{cat}$ ± SEM. **j** Immunoblot analysis of pervanadate-treated MCF10A lysates incubated with 0.2 μM PTPRK-D1 or 5.0 μM of PTPRK-D1.D1057E, PTPRU-D1 or PTPRU-D1.E1053D recombinant PTP domains for 16 h at 4 °C. Source data for (**c-e**) and (**g-j**) are provided as a Source Data file.

Substitution of the WPD-loop aspartate for glutamate, as seen in PTPRU, is common amongst the D2 pseudophosphatase domains of RPTPs (Supplementary Fig. 5a)[7]. Previously, PTP1B activity was reduced by several orders of magnitude upon mutation of the corresponding aspartate to glutamate (D181E)[38]. Similarly, mutating the WPD loop in PTPRK-D1 (D1057E) results in a ~115-fold reduction in enzymatic turnover versus WT, having low residual activity with a $k_{cat} = 0.0389\ s^{-1}$ (Fig. 1i). Critically, PTPRU-D1.E1053D, where the Glu has been reverted to the canonical Asp, remained inactive against cellular pTyr and pNPP (Fig. 1i, j). These data therefore suggest that an Asp to Glu substitution in the WPD loop alone is insufficient to account for complete loss of PTPRU-D1 enzymatic activity.

**Structure of PTPRU-D1.** In order to better understand the mechanisms underlying the lack of catalytic activity of PTPRU, we determined the structure of the D1 domain. The X-ray crystal structure of PTPRU-D1 was solved by molecular replacement using the PTPRK-D1 domain[39] (PDB ID: 2C7S) and refined to 1.72 Å resolution (Table 1). The overall fold of PTPRU-D1 closely resembles related phosphatase domains (RMSD of 1.0 Å² over 237 Cα atoms with PTPRK, Fig. 2a). However, the PTPRU structure reveals several key differences that likely contribute to its catalytic inactivity. The most striking difference is the absence of an ordered pTyr recognition loop (Fig. 2a). The structure and sequence of this loop is well conserved across the PTPs (Fig. 2b and Supplementary Fig. 5b), with the key tyrosine of the 'KNRY' motif creating an active site pocket deep enough to exclude pSer/Thr residues, whilst favourably stacking with the phenol ring of bound pTyr substrates (Fig. 2c). In the PTPRU crystal structure residues 904–925 encompassing the pTyr recognition loop region are disordered, resulting in almost complete loss of the pocket that would normally bind the pTyr substrate.

Another striking difference observed in our PTPRU-D1 structure was the conformation of the PTP loop containing the catalytic cysteine (Fig. 2d). The conformation of this loop is highly conserved in classical PTPs (Fig. 2di), but in PTPRU-D1 this loop is arranged such that the sidechain of T1089 is in close proximity to the active site cysteine (3.0 Å between T1089 OG1 and C1085 SG atoms, Fig. 2dii). The T1089 sidechain forms a hydrogen bond with the backbone amide of R1091 capping the adjacent helix (Supplementary Fig. 6a, b). A threonine at this position in the PTP loop is unique to the PTPRU-D1 domain and the more common hydrophobic residues at this position (Ala, Val, and Ile) are not capable of forming an equivalent interaction (Supplementary Fig. 1). This new loop orientation blocks the catalytic cysteine and would directly interfere with pTyr binding (Fig. 2e). The combined effect of a disordered pTyr recognition

loop and reorientation of the catalytic PTP loop is the loss of key structural features normally required for binding and processing of phosphorylated substrates (Fig. 2c, e).

An additional loop adjacent to the active site (C1121 to M1127) also adopts a conformation that differs from the canonical fold (Fig. 2dii lower left and Supplementary Fig. 6c). In all available structures, this loop is stabilised via hydrogen bonds between a conserved arginine and backbone carboxyl groups in this loop (R1119 in PTPRK, Supplementary Fig. 6d).

**Table 1 Data collection and refinement statistics.**

|  | PTPRU-D1 reduced | PTPRU-D1 oxidised |
|---|---|---|
| **Data collection** |  |  |
| Beamline | I04 | I03 |
| Wavelength (Å) | 0.9795 | 0.9796 |
| Space group | $C\ 2\ 2\ 2_1$ | $C\ 2\ 2\ 2_1$ |
| Cell dimensions |  |  |
| *a, b, c* (Å) | 61.8, 107.9, 88.3 | 62.1, 107.9, 89.3 |
| α, β, γ (°) | 90, 90, 90 | 90, 90, 90 |
| Resolution (Å) | 53.97–1.72 | 89.28–1.97 |
|  | (1.75–1.72)[a] | (2.00–1.97) |
| $R_{merge}$ | 0.148 (0.904) | 0.128 (1.061) |
| $R_{pim}$ | 0.043 (0.284) | 0.055 (0.458) |
| $CC_{1/2}$ | 0.998 (0.501) | 0.995 (0.527) |
| $I/σI$ | 9.7 (2.0) | 9.0 (1.6) |
| Completeness (%) | 100 (99.0) | 100 (100) |
| Redundancy | 12.9 (11.0) | 6.4 (6.3) |
| **Refinement** |  |  |
| Resolution (Å) | 53.97–1.72 | 46.19–1.97 |
|  | (1.78–1.72) | (2.07–1.97) |
| No. reflections | 31740 | 21577 |
| $R_{work}/R_{free}$ | 0.182/0.201 | 0.214/0.245 |
| No. atoms |  |  |
| Protein | 2020 | 1953 |
| Ligand/ion | 2 | 0 |
| Water | 117 | 94 |
| *B*-factors |  |  |
| Protein | 29.12 | 37.07 |
| Ligand/ion | 31.41 | – |
| Water | 33.87 | 43.17 |
| Ramachandran |  |  |
| Favoured (%) | 96.4 | 98.3 |
| Outliers (%) | 0.0 | 0.0 |
| R.m.s. deviations |  |  |
| Bond lengths (Å) | 0.012 | 0.005 |
| Bond angles (°) | 1.110 | 0.863 |

[a]Values in parentheses are for highest-resolution shell.

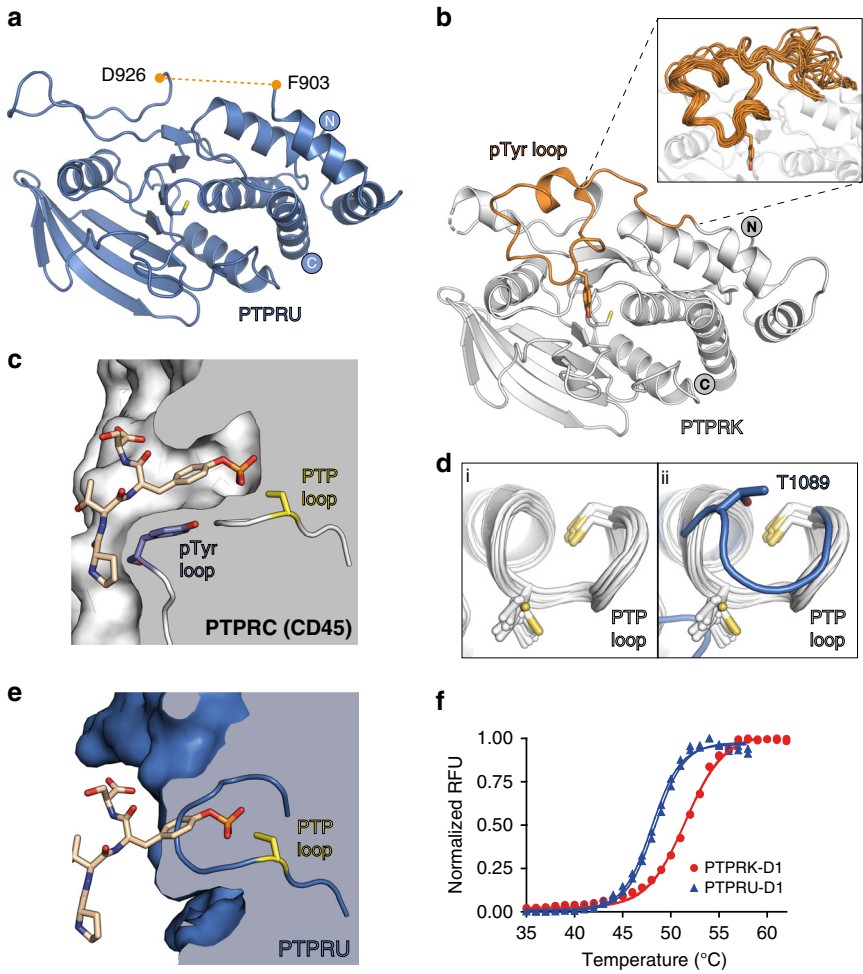

**Fig. 2 The structure of PTPRU-D1 reveals conformational rearrangements in key catalytic motifs. a** Ribbon diagram showing the overall structure of the PTPRU-D1 domain (blue). The boundaries of the pTyr recognition loop, which is disordered in the structure, are labelled and indicated by a dashed line (orange). **b** Ribbon diagram showing the overall structure of the PTPRK-D1 domain (white; PDB ID: 2C7S). The pTyr recognition loop is highlighted in orange with Y917 of the 'KNRY' motif shown in stick representation. Inset: Ribbon diagram showing structural alignment of the pTyr recognition loop for all classical PTP domains with a solved structure (28 total). Details of PTPs and corresponding PDB IDs are outlined in Supplementary Table 2. **c** Cross-section of the PTPRC (CD45) D1 domain in complex with a bound pTyr peptide (PDB ID: 1YGU). The key tyrosine of the pTyr recognition loop 'KNRY' motif (violet sticks) interacts with the pTyr phenol ring (wheat coloured sticks) and creates a deep pocket allowing access to the catalytic cysteine of the PTP loop (yellow sticks). **d** (i) Ribbon diagram showing structural alignment of the PTP loop region for all classical PTP domains with a solved structure (28 total). Details of PTPs and corresponding PDB IDs are outlined in Supplementary Table 2. (ii) Overlay of the PTPRU-D1 PTP-loop (blue), which exhibits a distinct conformation. The unique T1089 of PTPRU and the corresponding residue in the aligned structures, as well as the catalytic cysteine are shown in stick representation. **e** Cross-section following superposition of PTPRU-D1 (blue) onto the PTPRC-D1 in complex with a pTyr peptide (PDB ID: 1YGU) in the same orientation as in (**c**). Structural changes in the PTP loop block the active site, preventing access to the catalytic cysteine (yellow sticks). **f** Differential scanning fluorimetry thermal profiles of PTPRK-D1 (red) and PTPRU-D1 (blue) recombinant proteins. Source data for (**f**) is provided as a Source Data file.

This arginine is completely conserved in all D1 sequences except PTPRU where it is uniquely a cysteine (C1121, Supplementary Fig. 7). One consequence of this loss of an arginine is re-orientation of the nearby methionine (Supplementary Fig. 6c, d) and de-stabilisation of this loop. Residues C1121 to M1127 in PTPRU are not well ordered and challenging to build in a single, reliable conformation in the electron density suggesting it may adopt multiple conformations. In PTP1B (PTN1), the equivalent arginine (R254, Supplementary Fig. 7) has been suggested to form a secondary pTyr binding site via binding of a peptide containing tandem pTyr[40]. The importance of a cysteine residue in this secondary binding site, in a position resembling that of an active site cysteine, remains unclear. Consistent with the identified disorder in specific regions of the PTPRU-D1 structure, this domain has reduced thermal stability compared to PTPRK-D1, as

shown by a lower global melting temperature (PTPRU-D1 = 48.2 °C, PTPRK-D1 = 51.5 °C; Fig. 2f).

**Role of PTPRU motifs in activity and stability.** To investigate the consequence of the observed structural rearrangements in the PTPRU PTP and pTyr recognition loops, we generated a series of point mutations in key residues, as well as chimeric D1 domains harbouring reciprocal substitutions of the PTPRU and PTPRK pTyr recognition loops (Fig. 3a). In dephosphorylation assays, introduction of the unique T1089 of PTPRU in to PTPRK (A1093T) was not sufficient to inactivate PTPRK-D1 (Fig. 3b). Further, removal of the active site threonine was insufficient to reactivate PTPRU-D1 and a tandem E1053D and T1089A also remained inactive (Fig. 3b). Introduction of the highly divergent

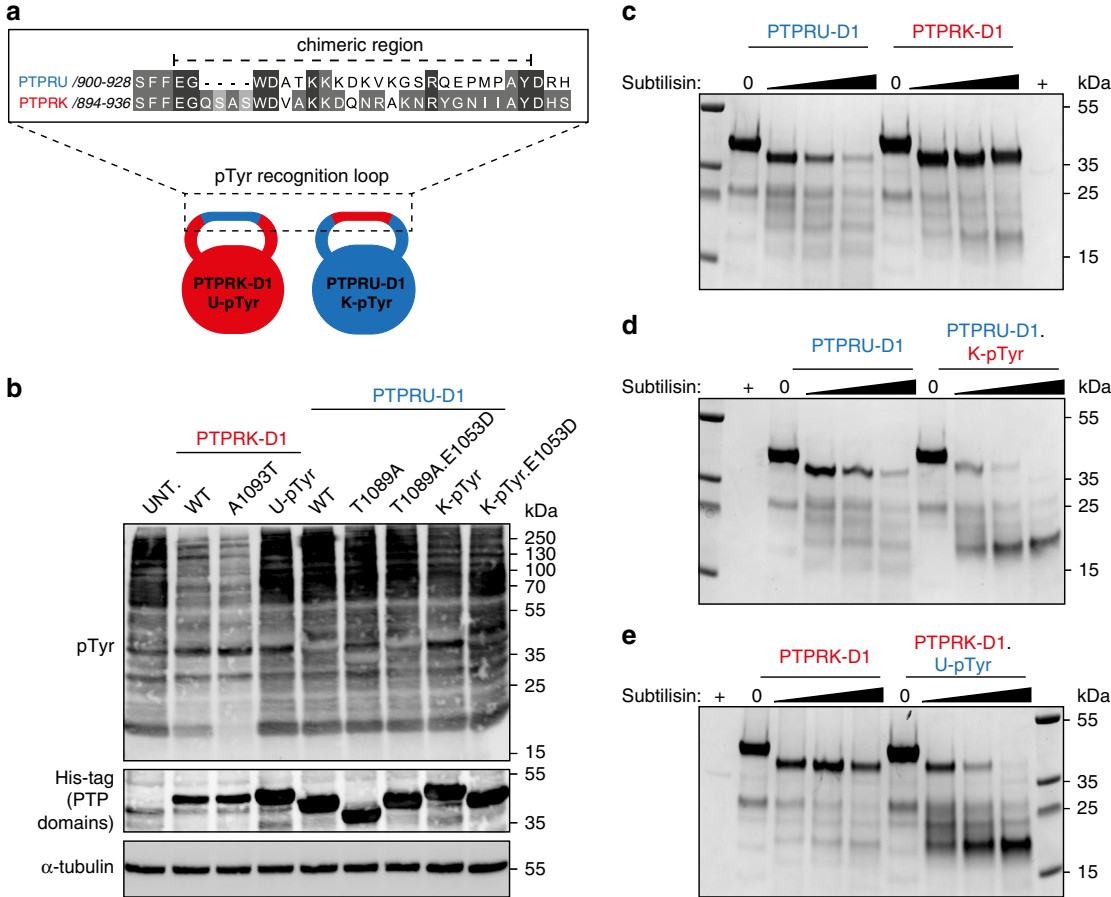

**Fig. 3 The role of PTPRU motifs in activity and stability. a** Schematic diagram of the pTyr recognition loop chimeric proteins used in this study. Inset: Sequence alignment of the pTyr recognition loops of PTPRU and PTPRK. The substituted sequence for chimeric proteins is highlighted (dashed line). **b** Immunoblot analysis of pervanadate-treated MCF10A lysates incubated with either 0.2 μM PTPRK-D1, PTPRK-D1.A1093T or 5 μM PTPRK-D1.U-pTyr, PTPRU-D1, PTPRU-D1.T1089A, PTPRU-D1.E1053D.T1089A, PTPRU-D1.K-pTyr, PTPRU-D1.K-pTyr.E1053D recombinant proteins. NB: All PTP domains are tagged with a 6xHis.TEV.Avi-tag (See "Methods") apart from PTPRU-D1.T1089A which has a 6xHis-tag only, resulting in lower apparent $M_W$. **c-e** Limited proteolysis using subtilisin (0, 0.625, 1.25 and 2.5 μg) of (**c**) PTPRU-D1 and PTPRK-D1, (**d**) PTPRU-D1 and PTPRU-D1.K-pTyr, (**e**) PTPRK-D1 and PTPRK-D1. U-pTyr followed by SDS-PAGE and Coomassie staining. Control lanes showing 2.5 μg subtilisin protein alone are included (+). Source data for (**b-e**) are provided as a Source Data file.

PTPRU-D1 pTyr recognition loop in to PTPRK-D1 results in loss of activity, however introduction of this loop from PTPRK in to PTPRU-D1 does not restore activity (Fig. 3b). The inability of these mutations to induce PTPRU-D1 activity was also confirmed in pNPP activity assays (Supplementary Fig. 8). Furthermore, introduction of an E1053D mutation combined with the PTPRK pTyr recognition loop also does not result in any detectable PTPRU-D1 activity (Fig. 3b). Therefore, the inactivity of PTPRU-D1 cannot be explained simply in terms of any single change to the PTP, WPD or pTyr recognition loop.

To determine the role of the pTyr recognition loop in protein stability, we subjected WT and chimeric D1 domains to limited proteolysis with subtilisin. The PTPRU-D1 domain showed higher susceptibility to proteolytic cleavage than PTPRK-D1, as would be predicted due to the disorder of the pTyr recognition loop (Fig. 3c). Introduction of the PTPRK pTyr recognition loop into PTPRU-D1 does not confer greater resistance to proteolysis, suggesting that the PTPRK loop cannot adopt a folded conformation in the context of the PTPRU-D1 domain (Fig. 3d). Indeed, the PTPRK loop appears to further destabilise PTPRU-D1. Introduction of the PTPRU pTyr recognition loop into PTPRK-D1 does result in greater susceptibility to proteolytic cleavage, supporting that this loop is again unable to form a

folded, protease-resistant conformation (Fig. 3e). These results are consistent with proteolysis using trypsin protease, confirming that any change in cleavage is not caused by an altered number of proteolytic cleavage sites introduced when generating chimeric sequences (Supplementary Fig. 9).

**Structure of oxidised PTPRU-D1**. In an attempt to determine if substrate binding might induce folding of the pTyr recognition loop or rearrangement of the catalytic PTP loop, we soaked PTPRU-D1 crystals with several potential ligands including $PO_4$, pTyr, and PNP. In none of the datasets collected for these crystal soaks was there any evidence of ligand binding in the active site or any induced folding of the pTyr recognition loop. However, these crystals, collected 4 weeks after the initial datasets, had clearly undergone oxidation resulting in the formation of a disulphide bridge between the highly conserved catalytic C1085 and the vicinal "backdoor" C998 cysteines (Fig. 4ai and Supplementary Fig. 2). This alternate conformation involving disulphide bond formation with nearby cysteines has been observed for several other related phosphatases (Fig. 4aii)[41–44]. Disulphide bond formation has been proposed to protect the catalytic cysteine from oxidative damage and/or function as a redox-

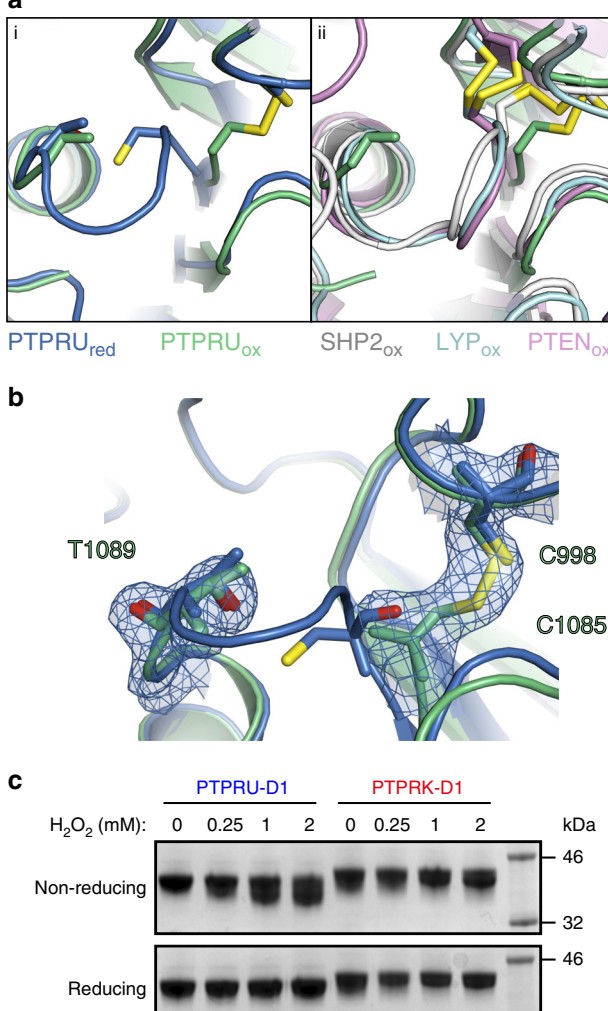

**a**

(i) PTPRU_red    PTPRU_ox

(ii) SHP2_ox    LYP_ox    PTEN_ox

**b**

T1089    C998    C1085

**c**

PTPRU-D1          PTPRK-D1

H$_2$O$_2$ (mM):  0  0.25  1  2    0  0.25  1  2    kDa

Non-reducing                                        — 46

                                                    — 32

Reducing                                            — 46

                                                    — 32

**Fig. 4 The PTPRU D1 domain forms an intramolecular "backdoor" disulphide bond. a** (i) Ribbon diagram of the reduced PTPRU-D1 PTP-loop region (PTPRU_red, blue) overlaid with PTPRU-D1 oxidised (PTPRU_ox, green). (ii) Ribbon diagram of the PTPRU_ox structure (green) overlaid with previously solved "backdoor" disulphide containing structures (SHP2, white; LYP, cyan; PTEN, violet; PDB IDs 6ATD, 3H2X and 5BUG respectively). **b** Reduced and oxidised models of PTPRU-D1 coloured as in **a** (i), showing electron density ($2F_o - F_c$ contoured at 0.8 e$^-$/Å$^3$, blue) for the "backdoor" disulphide and PTP-loop of the oxidised form. **c** In vitro oxidation of PTPRU and PTPRK D1 domains by hydrogen peroxide followed by reducing and non-reducing SDS-PAGE and Coomassie staining. Source data for (**c**) is provided as a Source Data file.

sensitive mechanism for reversible PTP inactivation[41–44]. The formation of this disulphide in PTPRU-D1 destabilises the conformation of adjacent residues in the PTP-loop as there is no clear density in which to model S1086-G1088 (Fig. 4b). The nearby loop (C1121-M1127) described previously as reoriented in the reduced form, is further destabilised in this oxidised structure as evidenced by a lack of electron density for this region. Non-reducing SDS-PAGE of PTPRU-D1 recombinant protein in the presence of hydrogen peroxide results in a mobility shift consistent with disulphide formation in solution, which is completely reversed under reducing conditions (Fig. 4c). While PTPRK-D1 conserves this "backdoor" cysteine, it does not undergo detectable disulphide formation under the same conditions (Fig. 4c). Thus,

the catalytic cysteine of PTPRU-D1 can undergo reversible oxidation, involving intramolecular disulphide formation, identifying this domain as a redox-sensitive pseudophosphatase.

**PTPRU interacts with PTPRK substrates.** To understand the role of PTPRU in signalling, we exploited our previously reported observations based on D1 and D2 domain-swapping chimeras showing that the PTPRK, but not PTPRM, D2 domain was critical for recognition of Afadin[3], a reported PTPRU interactor[30]. We generated an in vivo biotinylated chimera consisting of the active PTPRK-D1 and the PTPRU-D2 domain (Fig. 5a and Supplementary Fig. 10). We then tested the ability of chimeric proteins to bind and dephosphorylate PTPRK substrates. To probe protein-substrate interactions, we conjugated biotinylated chimeric proteins to streptavidin beads for in vitro pull-downs from pervanadate-treated cell lysates followed by immunoblotting. While the PTPRK and PTPRM substrate p120-Catenin (p120$^{Cat}$)[3,45] could interact with all chimeras regardless of D2 domain, we found that unlike PTPRM-D2, the PTPRU-D2 domain is sufficient for binding to Afadin (Fig. 5b). Consistent with this interaction data, immunoprecipitation of tyrosine phosphorylated proteins from dephosphorylation assays confirmed the PTPRU-D2 domain as being sufficient to recruit Afadin for dephosphorylation by the active PTPRK-D1 domain (Fig. 5c). Our data suggest that in cells PTPRU will bind but not dephosphorylate PTPRK substrates.

Previously we have identified specific p120$^{Cat}$ pTyr residues (Y228, Y904) which are dephosphorylated by PTPRK and PTPRM D1 domains, and are hyperphosphorylated in PTPRK-KO cells[3]. We confirmed by in-lysate dephosphorylation assays that while all PTPRK-D1-containing chimeras dephosphorylate pY228 and pY904, PTPRU-D1 cannot dephosphorylate these p120$^{Cat}$ sites (Fig. 5d, e). To investigate the cellular consequence of PTPRU binding to PTPRK substrates, we generated CRISPR-Cas9 mediated PTPRU-KO MCF10A cells. As expected, we were able to observe hyperphosphorylation of p120$^{Cat}$-pY228 and -pY904 in PTPRK-KO cells vs WT (Fig. 5f). Strikingly, deleting PTPRU resulted in hypophosphorylation of both p120$^{Cat}$-pY228 and -pY904 vs WT levels (Fig. 5f, g). Taking our interaction and dephosphorylation data together, this supports a mechanism in which PTPRU can bind substrates and protect them from dephosphorylation by related phosphatases.

## Discussion

The receptor PTPRU possesses several sequence variants in key catalytic motifs including a highly divergent pTyr recognition loop sequence, a unique Thr within the PTP loop and a Glu substitution in the WPD loop. Here we show that PTPRU does not exhibit detectable phosphatase activity against a range of substrates or in pTyr dephosphorylation assays with cell lysates. Our structural data identify multiple features that would disrupt both pTyr binding and catalytic activity. Despite its inactivity, we demonstrate that PTPRU can bind to key proteins previously reported as substrates for its catalytically active paralog PTPRK. This supports a role for PTPRU as a scaffold that competes with active phosphatases at the plasma membrane to locally influence tyrosine phosphorylation dynamics and potentially cell–cell adhesion.

Previous studies have shown that WPD to WPE mutations in several active phosphatases results in significant reduction in enzyme activity[13,38]. We show a similar substantial decrease in PTPRK activity following the introduction of the WPE sequence change. However, mutation of the PTPRU WPD loop to the canonical sequence (E1053D) was not sufficient to rescue any detectable phosphatase activity. Our structure of the PTPRU-D1

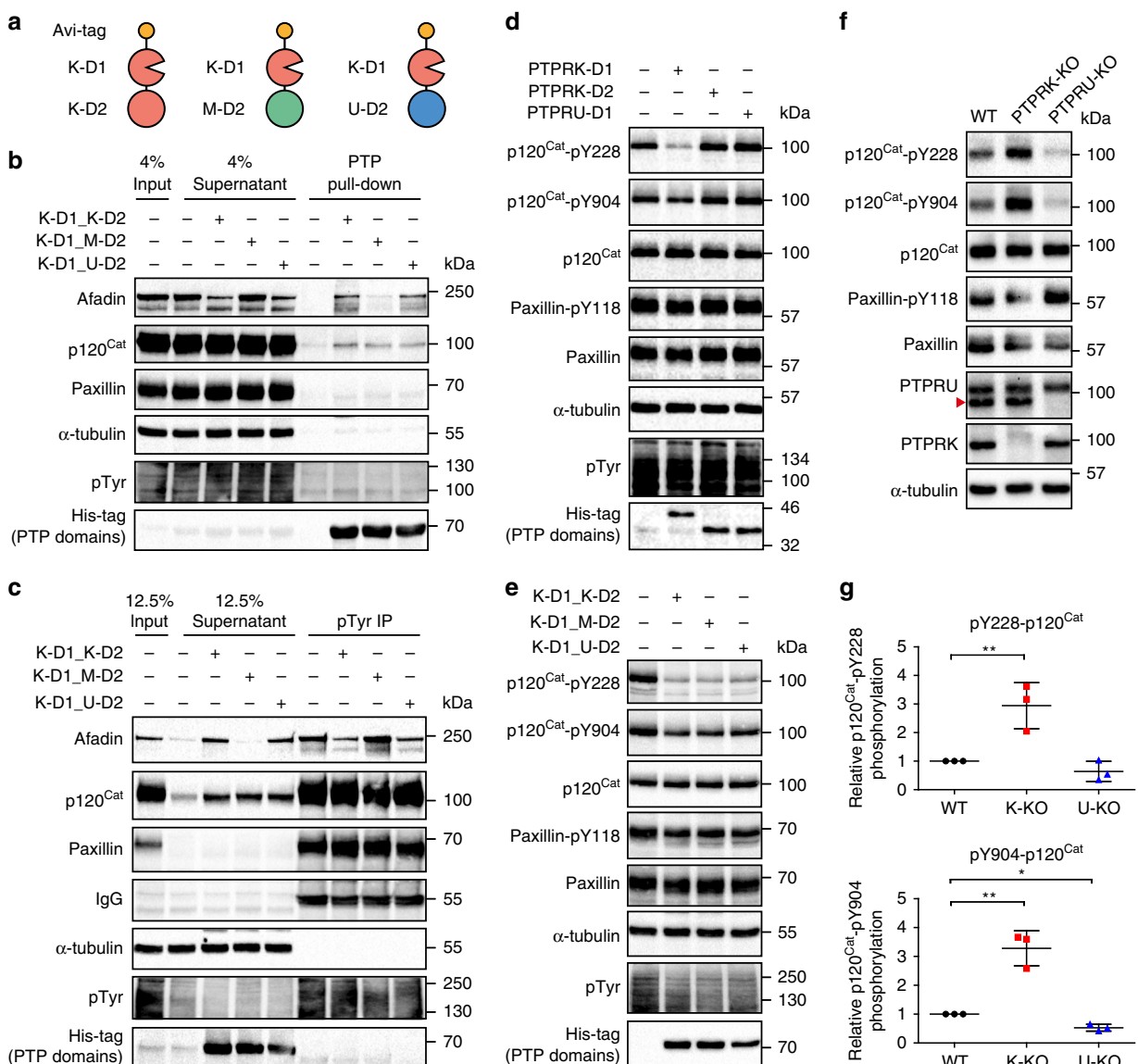

**Fig. 5 PTPRU binds but does not dephosphorylate PTPRK cellular substrates. a** Schematic diagram of chimeric tandem PTP domains used in this study. **b** Immunoblot analysis of recombinant protein pull-downs from pervanadate-treated MCF10A lysates, using the indicated chimeric tandem PTP domains. **c** Immunoblot analysis of pTyr immunoprecipitations (IPs) from dephosphorylation assays. Pervanadate-treated MCF10A lysates incubated for 1.5 h at 4 °C with 300 nM of the indicated chimeric tandem PTP domains. Dephosphorylated proteins are depleted from IPs and enriched in supernatants. **d** Immunoblot analysis of pervanadate-treated MCF10A lysates incubated for 1.5 h at 4 °C with 100 nM of PTPRK-D1, PTPRK-D2 or PTPRU-D1 recombinant domains. **e** Immunoblot analysis of pervanadate-treated MCF10A lysates incubated for 1.5 h at 4 °C with 300 nM of the indicated chimeric tandem PTP domains. **f** Immunoblot analysis of confluent WT, PTPRK-KO, and PTPRU-KO MCF10A cell lysates, harvested 24 h post media change. **g** Densitometry of p120[Cat] phosphorylation levels from (**f**), normalised against total p120[Cat] protein levels. Error bars represent ±SEM of $n = 3$ independent experiments. Unpaired student's $t$-test: $*p < 0.05$, $**p < 0.005$. Source data for (**b**–**g**) are provided as a Source Data file.

reveals that there are two key structural changes within this domain that alter the pTyr binding pocket and therefore are likely to explain the lack of phosphatase activity. The first is the disordered pTyr recognition loop. The absence of electron density for this loop was unexpected as, although the PTPRU sequence is highly divergent, it does retain a conserved arginine (R918) that in related structures binds back into the main PTP fold and interacts with residues that are conserved in PTPRU-D1. Not only does loss of this ordered loop drastically alter the shape of the pTyr binding pocket, it also contributes to decreased protein stability as demonstrated by the increased susceptibility of PTPRU to proteolysis and its lower melting temperature relative

to PTPRK-D1. The second key structural change in PTPRU-D1 relates to the catalytic PTP loop, which has undergone a substantial rearrangement resulting in the occlusion of the pTyr binding site, blocking the catalytic cysteine. Having identified that the sequence changes in PTPRU cause significant structural rearrangements to key loops required for enzyme activity we generated a series of point mutations and chimeric constructs to test their individual and combined effects on phosphatase activity. Using chimeric proteins where the pTyr loops of PTPRU and PTPRK were exchanged, we showed that although the PTPRU pTyr sequence was sufficient to inactivate PTPRK, the canonical pTyr loop sequence did not result in any phosphatase activity for

the PTPRU chimera. Furthermore, the PTPRU-D1 K-pTyr chimera is even more destabilised, as shown by increased susceptibility to proteolysis, suggesting this loop cannot fold correctly to form a pTyr-binding pocket as seen for PTPRK and other PTPs. Despite lacking a pTyr-binding pocket, it remained a possibility that PTPRU could process phosphorylated substrates as it retains the catalytic cysteine. Therefore, we tested the impact on phosphatase activity of the Thr (T1089) that blocks this cysteine in the PTP loop. Interestingly, the introduction of a Thr into the equivalent position of the PTPRK-D1 does not reduce its phosphatase activity while mutation of the PTPRU Thr to Ala does not induce detectable PTPRU activity. Furthermore, combinations of multiple mutations that restore canonical PTP sequences to PTPRU including E1053D and T1089A or replacing the pTyr loop and E1053D were still unable to rescue any activity. This combination of biochemical and structural analysis demonstrates that there are multiple mechanisms contributing to the absence of phosphatase activity in PTPRU-D1.

An intriguing feature of RPTP family inactive D2 pseudo-phosphatase domains is the retention of the catalytic cysteine residue, as seen in PTPRU (Supplementary Fig. 2). The conservation of this residue in inactive domains raises the question of whether it plays an alternative, non-catalytic role. Our structure of oxidised PTPRU-D1 demonstrates that this cysteine has the capacity to form a disulphide bond with a "backdoor" cysteine similar to that seen in the active SHP2, LYP, and PTEN phosphatases[41-43]. In these enzymes, the formation of a disulphide bond has been attributed to the need to protect the catalytic cysteine from irreversible oxidative damage or to allow reversible redox-sensitive inactivation[46,47]. Our observation here of a similar intramolecular disulphide in an inactive pseudophosphatase domain suggests that the proposed roles for the disulphide formation may extend beyond the modulation of enzyme activity. Importantly, oxidation of PTPRU and RPTP domains has been reported in cells[48]. Previous studies on the PTPRA D2 pseudophosphatase domain suggest oxidation can promote an intermolecular disulphide bond[18], or a conformational change that is translated to the extracellular domain[49]. Thus, for several PTP domains catalysis is impaired yet redox sensing is preserved and warrants further investigation.

In addition to promoting a "backdoor" disulphide, PTP oxidation can induce chemical modification of the catalytic cysteine. One such modification is the formation of a sulfenyl-amide intermediate, as demonstrated for PTP1B[50,51]. This modification involves the sidechain of the catalytic cysteine forming a covalent link to the backbone nitrogen of an adjacent residue, resulting in a substantial change to the conformation of the catalytic PTP loop. Interestingly, this loop conformation is highly similar to that seen in PTPRU-D1. In PTP1B this renders the enzyme inactive but is reversible upon reduction and is proposed to be a protective intermediate during redox-regulated inhibition. This conformation of the PTPRU-D1 PTP loop is not induced by oxidation, it is instead present in the reduced form rendering the enzyme unable to bind pTyr. This suggests PTPRU has evolved to adopt an inactive conformation, even under reducing conditions.

The ICDs of other members of the R2B family comprise an active membrane-proximal D1 domain and an inactive membrane-distal D2 domain. Despite the sequence divergence of PTPRU-D1 from its paralogs (Supplementary Fig. 1) and its lack of catalytic activity, this domain still possesses higher sequence identity to D1 domains than to D2 domains (69% sequence identity with R2B family D1 domains, 28% identity with R2B family D2 domains; Supplementary Table 1). Therefore, PTPRU retains a bona-fide D1 and D2 domain topology similar to that of related enzymes but with a D1 domain that has diverged to be catalytically inactive. By using chimeric ICDs containing D1 and D2 domains from PTPRU, PTPRK, and PTPRM in cell-based dephosphorylation assays we show that the D2 domain of PTPRU can recruit substrates for dephosphorylation by the active D1 domain of PTPRK. We previously showed that protein binding to the ICD of the related receptor PTPRK was phosphorylation-independent[3], suggesting that the exclusion of pTyr by the occluded PTPRU-D1 active site would not necessarily inhibit binding of protein interactors in the context of the full PTPRU-ICD, consistent with what we observe. This ability to bind substrates that overlap with active phosphatases, combined with the lack of phosphatase activity of PTPRU suggests that the likely role of PTPRU in cells is to act as a decoy receptor that sequesters substrates protecting them from dephosphorylation. In support of this, we find that genetic deletion of PTPRU leads to a reduction in phosphorylation levels of the PTPRK substrate p120[Cat]. In this way, PTPRU may modify cell signalling by altering the rate or extent of tyrosine dephosphorylation by related, active RPTP family members. The absence of phosphatase activity demonstrated here for PTPRU does not diminish its importance but highlights a new pseudophosphatase function in cell signalling.

## Methods

**Plasmids and constructs.** Amino acid (aa) numbering used throughout is based on the following sequences; PTPRU; UniProt ID: Q92729-1; PTPRK; UniProt ID: Q15262-3; PTPRM; UniProt ID: P28827-1; PTPRT; UniProt ID: O14522-3. For bacterial expression, the human cDNA sequence encoding PTPRU-D1 (aa. 871-1153) was subcloned into a modified pET-15b expression vector in frame with an N-terminal hexahistidine-tag followed by a TEV protease recognition site and Avi-tag[3] (His6.TEV.Avi-tag). Chimeric D1 + D2 constructs were generated by subcloning of PTPRK-D1(aa 865–1153).BstBI.PTPRU-D2(aa 1150–1446) in to pET-15b.His6. TEV.AviTag. pET-15b.His6.TEV.Avi vectors expressing PTPRK-D1, PTPRK-D2, PTPRK-D1.BstBI.PTPRK-D2, and PTPRK-D1.BstBI.PTPRM-D2 were generated in a previous study[3]. For crystallisation studies PTPRU-D1 was subcloned into a modified pET-15b expression vector in frame with an N-terminal His6 tag only. For expression in mammalian cells PTPRU-ICD (aa 904–925) and PTPRK-ICD (aa 776-1446) were subcloned in to a modified pmScarlet_C1 (Addgene #85042) with mScarlet replaced by a N-terminal 3xFLAG-tag. Chimeric pTyr recognition loop D1 domains were generated using a previously described E. coli homologous recombination method[52]; PTPRU-D1 and PTPRK-D1 bacterial expression constructs were PCR linearised internally to lack a pTyr-loop (PTPRU, aa 904-925; PTPRK, aa 894–926) and combined with annealed oligonucleotides encoding the pTyr loop of the reciprocal domain, with 20 bp homology arms between each fragment. Mutations were introduced by site-directed mutagenesis using polymerase chain reaction with Phusion Hot Start II DNA polymerase. All primers and oligonucleotides used in this study are outlined in Supplementary Table 3.

**Antibodies.** The antibodies used for immunoblot analysis in this study are as follows. All antibodies were used at a dilution of 1:1000 in TBS-T (20 mM Tris-HCl, pH 7.6, 137 mM NaCl, 0.2% [v/v] Tween-20) with 3% (w/v) BSA unless otherwise indicated. Rabbit anti-pTyr (Cat#8954), mouse anti-HisTag (Cat#2366), rabbit anti-Paxillin (Cat#12065), rabbit anti-phospho-Paxillin (Y118; Cat#2541), rabbit anti-phospho-p120-Catenin (Y228; Cat#2911), and rabbit anti-phospho-p120-Catenin (Y904; Cat#2910) primary antibodies were all purchased from Cell Signalling Technology. Mouse anti-Afadin (Cat#610732) and mouse anti-p120-Catenin (Cat#610133) primary antibodies were purchased from BD Transduction Laboratories. Mouse anti-PTPRM (PTPRM cross-reactive; Cat#sc-56959)[21] primary antibody was purchased from Santa Cruz. Rabbit anti-PTPRK primary antibody was generated in a previous study[3]. Mouse anti-alpha-tubulin (Cat#T6199) and anti-FLAG (Cat#F7425) primary antibodies were purchased from Sigma Aldrich. HRP-conjugated anti-mouse (Cat#711-035-151) and anti-rabbit (Cat#711-035-152) secondary antibodies (1:5000 in TBS-T) were purchased from Jackson Immuno-Research.

**Protein expression and purification.** Escherichia coli BL21(DE3) Rosetta cells transformed with the relevant expression construct were cultured in 2X TY medium at 30 °C until $OD_{600} = 0.6$. Routinely, 1–2 mg of recombinant PTP was obtained per 1 L culture. Protein expression was induced with 1 mM isopropyl-thio-β-D-galactopyranoside for 18 h at 20 °C. For biotinylated Avi-tag constructs, 200 μM D-biotin (Sigma Aldrich) was added at the point of induction. After a freeze-thaw cycle, bacterial pellets were resuspended in ice-cold lysis buffer (50 mM Tris, pH 7.4 [PTPRK domains]/pH 8 [PTPRU domains], 500 mM NaCl, 5% glycerol, 0.5 mM TCEP) and lysed using high-pressure cell disruption (Constant Systems Ltd). Cell lysates were clarified by centrifugation at $40,000 \times g$ for 30 min. Cleared lysates were incubated with Ni-NTA agarose beads (Qiagen) for 1 h at 4 °C. Ni-NTA beads were packed in to a 10 ml gravity-flow column and equilibrated

with 10 bed volumes of purification buffer (for PTPRU constructs; 50 mM Tris-HCl, pH 8, 500 mM NaCl, 5% glycerol, 5 mM DTT, for PTPRK constructs; 50 mM Tris-HCl, pH 7.4, 150 mM NaCl, 5% glycerol, 5 mM DTT) containing 5 mM imidazole. Ni-NTA beads were then washed with 20 bed volumes of purification buffer containing 20 mM imidazole, followed by elution in purification buffer containing 250 mM imidazole. Eluted protein was further purified by size-exclusion chromatography on a HiLoad Superdex 200 pg 16/600 column (GE Healthcare) equilibrated in purification buffer. For crystallization-quality PTPRU-D1 domain, protein was buffer exchanged by iterative concentration/dilution in a 10 K MWCO centrifugal filter unit (Merck Millipore) against low-salt buffer (50 mM Tris-HCl, 10 mM NaCl, pH 8, 5% glycerol, 5 mM DTT) until a final NaCl concentration <15 mM. Protein was further purified by anion exchange chromatography on a MonoQ 5/50 GL column (GE Healthcare) equilibrated in low-salt buffer and bound protein was eluted by a linear 20 ml gradient against high-salt buffer (50 mM Tris-HCl, pH 8, 1 M NaCl, 5% glycerol, 5 mM DTT). Protein purity was assessed by SDS-PAGE and staining with Coomassie (Instant Blue, Expedeon).

**Crystallisation**. Crystallisation experiments were performed in 96-well nanolitre-scale sitting drops (200 nl of 9.6 mg/ml PTPRU-D1 plus 200 nl of precipitant) equilibrated at 20 °C against 80 μl reservoirs of precipitant. Diffraction quality crystals grew against a reservoir of 0.1 M Bicine, pH 9, 1 M lithium chloride, 20% (w/v) polyethylene glycol 6000. For the oxidised structure, PTPRU-D1 crystals were soaked in 1 mM pTyr (Sigma Aldrich) for 3 min. Crystals were cryoprotected in reservoir solution supplemented with 20% (v/v) glycerol and flash-cooled by plunging into liquid nitrogen.

**X-ray data collection and structure solution**. X-ray diffraction data were recorded at Diamond Light Source (DLS) beamlines I03 and I04. Datasets were collected at $\lambda = 0.9795$ Å. Diffraction datasets were indexed and integrated using the automated data processing pipeline available at DLS, implementing XIA2 DIALS for the reduced dataset and XIA2 3dii for the oxidized dataset[53] then scaled and merged using AIMLESS[54]. Resolution cut-off was determined by $CC_{1/2} > 0.5$ and $I/\sigma I > 1.5$. The initial structure was solved by molecular replacement using Phaser[55], with human PTPRK-D1[39] (PDB ID: 2C7S) as a search model. Further refinements were performed using COOT[56] and phenix.refine[57]. Graphical figures of the PTPRU-D1 structure were rendered in PyMOL (Schrödinger, LLC).

**pNPP phosphatase activity assay**. Recombinant PTP domains were made up to 50 μL volumes in a 96-well microplate format in assay buffer (50 mM Tris-HCl, pH 7.4, 150 mM NaCl, 5% glycerol, 5 mM DTT, 100 μg/ml BSA). Serial dilutions of pNPP substrate (0–40 mM, New England Biolabs) were performed in assay buffer. Reaction plates containing both enzyme and substrate dilutions were incubated at 30 °C for 30 min prior to addition and mixing of 50 μl pNPP substrate to initiate reactions. Product formation was monitored for 15 min at 30 °C by measuring absorption at 405 nm in a Spectramax M5 plate reader (Molecular Devices), followed by fitting to a 4-nitrophenol (Sigma) standard curve of known concentration. Data were fitted using linear regression in GraphPad Prism to determine initial enzymatic rates ($V_0$). $V_0$ values at various substrate concentrations were fitted using non-linear regression and kinetic parameters ($V_{max}$ and $K_m$) calculated from the Michaelis-Menten equation in GraphPad Prism. $k_{cat}$ values were calculated using the equation $k_{cat} = V_{max}/[E_T]$.

**Cells and cell culture**. MCF10A and HEK-293T cells were obtained from the American Type Culture Collection (ATCC, CRL-10317 and CRL-3216 respectively). MCF10A cells were cultured in a previously described MCF10A growth medium[58] consisting of 1:1 DMEM:Ham's F12 supplemented with 5% (v/v) horse serum, 0.5 mg/ml hydrocortisone, 20 ng/ml EGF, 100 ng/ml cholera toxin and 10 mg/ml insulin. HEK-293T cells were cultured in DMEM supplemented with 10% (v/v) foetal calf serum. Cells were cultured at 37 °C in a humidified 5% $CO_2$ atmosphere in 75 cm[2] vented flasks.

**Flag immunoprecipitations and pNPP assay**. HEK-293T ($4 \times 10^6$) cells were reverse transfected in 10 cm[2] dishes with 8 μg expression constructs for 3xFLAG tagged ICDs using 24 μg polyethylenimine (PEI) in 500 μl OptiPRO serum free medium (Thermo Fisher Scientific). Cells were allowed to adhere and media replaced 16-h post-transfection.

For immunoprecipitation (IP), cells were transferred to ice 48-h post-transfection and washed twice with ice-cold 1X phosphate buffered saline (PBS). Cells were lysed in 700 μl ice-cold lysis buffer (50 mM Tris-HCl, pH 7.4, 150 mM NaCl, 10% [v/v] glycerol, 1% [v/v] Triton X-100, 1 mM EDTA, 10 mM NaF, 1 mM PMSF, 1X EDTA-free protease inhibitor) on ice for 30 min, with periodic agitation. Lysates were cleared by centrifugation at 14,000 x g, 15 min at 4 °C and supernatants transferred to chilled tubes. Total protein concentration was quantified by bicinchoninic acid (BCA) assay. Lysates were then adjusted to a final concentration of 5 mM DTT, to prevent air oxidation. Equal amounts of total cell lysate for each sample (~2 mg) was combined with 10 μl (20 μl of 50% slurry) of washed FLAG-M2 magnetic beads (Sigma Aldrich) in a total volume of 1 ml made up in lysis buffer and incubated for 2 h at 4 °C with rotation. Beads were then collected on a magnetic stand and supernatants removed. Beads were then

resuspended and washed once with 1 ml lysis buffer and four times with lysis buffer containing 500 mM NaCl.

For pNPP assays, IPs were washed once with 1 ml assay buffer (50 mM Tris-HCl, pH 7.4, 150 mM NaCl, 5% [v/v] glycerol, 5 mM DTT), then resuspended in 500 μl assay buffer. Reactions were initiated by adding 500 μl assay buffer supplemented with 20 mM pNPP (10 mM final concentration), and reaction tubes placed horizontally on an orbital shaking platform at 30 °C, 220 RPM. At each timepoint, beads were thoroughly resuspended and 100 μl samples were immediately added to 50 μl 0.58 M NaOH solution to terminate the reaction. Timepoint supernatants were transferred to a microplate. After the final timepoint, product formation was determined by measuring absorption at 405 nm in a Spectramax M5 plate reader (Molecular Devices), followed by fitting to a 4-nitrophenol (Sigma) standard curve. After assay completion, remaining beads were collected by magnet and washed in 1 ml TBS. Beads were then resuspended in 40 μl of TBS with 10 μl 5X SDS PAGE sample buffer and incubated at 95 °C for 10 mins. Beads were collected by magnet and supernatants used for SDS-PAGE and immunoblot analysis.

**BIOMOL green phosphatase assay**. Recombinant PTP domains were made up to 20 μl volumes in a 96-well microplate format in reaction buffer (50 mM Tris-HCl, pH 7.4, 150 mM NaCl, 5% glycerol, 5 mM DTT). Thirty microliters of 100 μM pSer, pThr, pTyr peptides (DADE-pY-LIPQQG and END-pY-INASL) or imido-diphosphate was added to protein wells to initiate reactions. Reactions were allowed to proceed for 15 min before termination by addition of 100 μl BIOMOL Green reagent (Enzo). Liberated phosphate was measured by absorbance at 360 nm in a Spectramax M5 plate reader (Molecular Devices), followed by interpolation to a standard curve of known phosphate concentration. Serial dilutions of phosphate were performed in reaction buffer using 800 μM phosphate standard (Enzo).

**Phosphoinositide phosphatase activity assay**. The commercially available Enzchek phosphate assay kit (Thermo Fisher Scientific) was used to measure the release of phosphate from the phosphoinositides lipids PI(4,5)P2 and PI(4)P. The kit was used according to the manufacturer's instructions. Briefly, PTPRU-D1 (3 μM) or the positive control PRL-3 (6 μM) were incubated in Enzchek reaction buffer containing 50 mm Tris-HCl pH 7.5, 2 mM MgCl2 plus 500 mM NaCl (only for PTPRU-D1) and 5 mM DTT. The reaction was initiated upon addition of each lipid substrate at a final concentration of 100 μM. The assay was conducted in triplicates at 37 °C in a BioTek Synergy H1 plate reader for 45 min and the release of phosphate was monitored measuring the absorbance at 360 nm over time. For every phosphoinositide substrate, a control without enzyme for blank subtraction was also measured. The data are represented as mean $+/-$ SD.

**Preparation of sodium pervanadate**. To create a 50 mM pervanadate stock solution, 30% $H_2O_2$ was first diluted to 0.3% $H_2O_2$ in 20 mM HEPES, pH 7.3. Fifty microliters of dilute $H_2O_2$ added to 490 μl 100 mM sodium orthovanadate (Alfa Aesar) and 450 μl $H_2O$, then mixed by gentle inversion and incubated at RT for 5 min. After incubation, excess $H_2O_2$ was quenched by the addition of a small amount of catalase (using 200 μl pipette tip) and mixed by gentle inversion. Pervanadate was freshly prepared and used immediately to avoid decomposition.

**Generation of pervanadate-treated lysates**. $3 \times 10^6$ MCF10A cells were seeded in 10 cm dishes and cultured to complete confluence (4 days). Media was then aspirated and cells treated with 8 ml of fresh growth medium containing 1 mM sodium pervanadate for 30 min at 37 °C, 5% $CO_2$. Cells were placed on ice and washed twice with ice-cold PBS. Cells were lysed in 600 μl per 10 cm dish of ice-cold lysis buffer (50 mM Tris-HCl, pH 7.4, 150 mM NaCl, 10% [v/v] glycerol, 1% [v/v] Triton X-100, 1 mM EDTA, 5 mM iodoacetamide [IAA], 1 mM sodium orthovanadate, 10 mM NaF, 1X EDTA-free protease inhibitor) by orbital shaking on ice, in the dark, at 4 °C for 30 min. Lysates were collected by cell-scraping and treated with 10 mM DTT on ice for 15 min. Note: EDTA was used to chelate excess vanadate and IAA for alkylation of PTP active site cysteines. Lysates were then cleared by centrifugation at 14,000 × g, 15 min at 4 °C. Supernatants were transferred to a fresh tube and snap frozen in liquid $N_2$ for storage at −80 °C until use.

**In lysate dephosphorylation assays**. Recombinant PTP domains (0–5 μM final concentration, as indicated) were mixed with 50 μl (800 μg) of freshly thawed pervanadate-treated cell lysate in a final volume of 400 μl made up in ice-cold 150 mM wash buffer with 5 mM DTT (20 mM Tris-HCl, pH 7.4, 150 mM NaCl, 10% [v/v] glycerol, 1% [v/v] Triton X-100, 1 mM EDTA, 5 mM DTT). Reactions were then incubated with rotation to complete dephosphorylation for 24 h at 4 °C unless stated otherwise. For time-course experiments, at each timepoint 72 μl of reaction was removed and added to 1.5 μl 20% (w/v) SDS (0.4% [w/v] SDS final) to stop reactions. Twenty-six microliters (~50 μg) of each sample was then mixed with 6.5 μl 5X sample loading buffer and stored at −20 °C prior to SDS-PAGE and immunoblot analysis.

**SDS-PAGE and immunoblotting**. Total protein concentrations of cell-lysates were quantified by BCA assay and 25–50 μg of lysate mixed in an appropriate volume of 5X sample loading buffer (0.25 M Tris-HCl, pH 6.8, 10% [w/v] SDS, 20% [v/v] glycerol, 0.1% [w/v] bromophenol blue, 10% [v/v] β-mercaptoethanol). Samples

were then incubated at 92 °C for 5 min and resolved by SDS-PAGE on a 10% resolving gel. Total protein was transferred to 0.2 µm reinforced nitrocellulose membranes by wet transfer and blocked in 5% (w/v) skimmed-milk in TBS with 0.2% TWEEN-20 (TBS-T; 20 mM Tris-HCl, pH 7.6, 137 mM NaCl, 0.2% [v/v] Tween-20) for 20–60 min. Membranes were then rinsed once with TBS-T before incubation with appropriate primary antibody in 3% (w/v) BSA/TBS-T overnight at 4 °C. Membranes were washed 3 × 10 min in TBS-T and then incubated with the appropriate species-specific HRP-conjugated anti-IgG secondary antibody for 1 h at RT. Following 3 × 10 min washes in TBS-T, membranes were developed using EZ-ECL solution (Geneflow) and imaged using a BioRad ChemiDoc MP imaging system (Bio-Rad). 2D-densitometry for quantification was carried out in Fiji[59].

**Limited proteolysis.** All steps were performed on ice unless otherwise stated. Trypsin and subtilisin proteases were reconstituted at 1 mg/ml in proteolysis buffer (50 mM Tris-HCl, pH 7.4, 150 mM NaCl, 10% glycerol, 1% Triton X-100, 1 mM EDTA, 5 mM DTT). Four micrograms of recombinant protein was incubated with 0, 0.625, 1.25, 2.5 µg subtilisin or 0, 1.25, 2.5 or 5 µg trypsin in a total volume of 25 µl made up in proteolysis buffer. Reactions were incubated on ice for 30 min and terminated by the addition of 10 µl 5X sample loading buffer. Samples were immediately incubated at 92 °C for 5 min and resolved by SDS-PAGE on a 16% SDS-polyacrylamide gel. Proteins were visualized by Coomassie staining and gels were imaged using a ChemiDoc MP imager.

**Differential scanning fluorimetry.** Differential scanning fluorimetry was performed using Protein Thermal Shift Dye kit (Thermo Fisher Scientific) as per manufacturer's protocol in a ViiA-7 Real-time PCR system (Thermo Fisher Scientific). Reaction mixes consisting of 2 µg recombinant protein and 1X Protein Thermal Shift Dye were made up to a total volume of 20 µl in purification buffer. Samples were then heated on a 1 °C/s gradient from 25 to 95 °C and protein unfolding at each temperature monitored by measurement of fluorescence at 580/623 nm (excitation/emission). Fluorescent signal vs. temperature was fitted to a non-linear Boltzman-sigmoidal regression in Graphpad Prism, with the $T_m$ calculated from the inflection point of the fitted curve.

**Reversible oxidation of recombinant protein.** All steps were performed on ice unless otherwise stated. Ten micrograms of recombinant protein was mixed with either 0, 0.25, 1 or 2 mM $H_2O_2$ in a total reaction volume of 50 µl in ice-cold oxidation buffer (50 mM Tris-HCl, pH 7.4, 150 mM NaCl, 5% glycerol). Reactions were then incubated on ice for 30 min. Each reaction was then split equally (25 µl, 5 µg) and incubated at 95 °C for 5 min with 7.5 µl 5X sample loading buffer with (reducing) or without (non-reducing) β-mercaptoethanol. Reduced and non-reduced samples were then resolved by SDS-PAGE on separate NuPage 4–12% bis-tris gels using 1X MOPS running buffer and visualized by staining with Coomassie. Gels imaged using a ChemiDoc MP imager.

**Assessment of recombinant protein biotinylation.** Ten micrograms of in vivo biotinylated recombinant protein was boiled for 5 min at 95 °C in an appropriate volume of 5X sample loading buffer. Samples were cooled to RT before addition of a 3-fold molar excess of streptavidin and incubated for 5 min at RT. Protein was then resolved by SDS-PAGE on a NuPAGE 4–12% Bis-Tris gel and visualized by staining with Coomassie stain. After destaining with ddH₂O, gels were imaged on a ChemiDoc MP imager and total levels of biotinylated protein quantified by 2D-densitometry in Fiji[59].

**Recombinant protein pull downs.** 50 µg of biotinylated Avi-tag recombinant PTP domains were bound to 167 µl of streptavidin-coated magnetic beads (New England Biolabs) made up to a total volume of 500 µl in ice-cold purification buffer (50 mM Tris, pH 7.4, 150 mM NaCl, 5% [v/v] glycerol, 5 mM DTT) at 4 °C for 1.5 h with rotation. Beads were collected using a magnetic stand and washed 3 times in ice-cold purification buffer, followed by two washes with ice-cold 150 mM NaCl wash buffer (20 mM Tris-HCl, pH 7.4, 150 mM NaCl, 10% [v/v] glycerol, 1% [v/v] Triton X-100, 1 mM EDTA). Bead conjugated PTP domains were then blocked in ice-cold 5% (w/v) BSA/150 mM NaCl wash buffer at 4 °C for 1 h with rotation. Pervanadate-treated cell lysates were pre-cleared by incubation with streptavidin magnetic beads (0.67 mg beads per ml of lysate) at 4 °C for 1 h with rotation. Blocked PTP domains were collected on a magnetic stand and washed twice with ice-cold 150 mM NaCl wash buffer. One milligram of pre-cleared pervanadate-treated lysate (250 µl) was incubated with PTP domain-bound beads in a total volume of 1 ml 150 mM NaCl wash buffer at 4 °C for 1.5 h with rotation. At 4 °C, beads were collected on a magnetic stand and supernatants removed. Bead bound protein was then washed twice by resuspension in ice-cold 150 mM NaCl wash buffer, followed by one wash in ice-cold 500 mM NaCl wash buffer (20 mM Tris-HCl, pH 7.4, 500 mM NaCl, 10% [v/v], 1% [v/v] Triton X-100, 1 mM EDTA) with no resuspension. Beads were then washed twice by resuspension in ice-cold 500 mM NaCl wash buffer followed by a final wash in ice-cold TBS (20 mM Tris-HCl, pH 7.4, 137 mM NaCl). Beads were resuspended in 15 µl TBS with 25 µl 5X sample loading buffer containing 2 mM biotin and incubated at 95 °C for 10 min. Beads and supernatants were stored at −20 °C prior to SDS-PAGE and immunoblot analysis.

**pTyr immunoprecipitation.** In total, 400 µl dephosphorylation reactions (prepared as described above) were diluted to a total volume of 800 µl (0.2% [w/v] SDS final concentration) in 150 mM NaCl wash buffer (20 mM Tris-HCl, pH 7.4, 150 mM NaCl, 10% [v/v] glycerol, 1% [v/v] Triton X-100, 1 mM EDTA). Five microliters of rabbit anti-pTyr antibody (Cell Signalling Technology) was added to each sample and incubation for 3 h at 4 °C with rotation. Immunoprecipitation was carried out by addition of 40 µl of washed protein G agarose beads and samples were incubated overnight (16 h) at 4 °C with rotation. Beads were collected by centrifugation at 15,000 × g for 30 s at 4 °C and washed five times with 1 ml of ice-cold 150 mM wash buffer. After washing, beads were resuspended in 2.5X sample loading buffer and incubated at 95 °C for 10 min. Supernatants were collected and stored at −20 °C prior to SDS-PAGE and immunoblot analysis.

**CRISPR/Cas9 genome editing.** Oligonucleotides encoding single guide RNAs (sgRNAs) targeting human *PTPRU* exon 1 and exon 14 were cloned into pSpCas9. (BB).mCherry and pSpCas9.(BB).eGFP respectively as previously described[60]. MCF10A cells were co-transfected with both plasmids by reverse transfection using Lipofectamine LTX/PLUS reagent as per manufacturer's instructions. Clonal cell-lines were established by single-cell sorting of mCherry-eGFP double positive cells by flow cytometry, 48-h post-transfection. After expansion of clones, PTPRU-KO clones were identified using immunoblot for PTPRU. sgRNA target sites were amplified from genomic DNA to confirm editing. Three independent confirmed PTPRU-KO clones were pooled to establish the final PTPRU-KO MCF10A population. PTPRK-KO MCF10A cells were generated in a previous study[3].

**Protein sequence alignments.** Protein multiple sequence alignments were generated using Clustal Omega[61] and edited using Jalview[62].

**Reporting summary.** Further information on research design is available in the Nature Research Reporting Summary linked to this article.

## Data availability

The atomic coordinates and structure factors have been deposited in the Protein Data Bank, www.pdb.org under accession codes 6SUB[https://doi.org/10.2210/pdb6SUB/pdb] (PTPRU-D1 reduced form) and 6SUC[https://doi.org/10.2210/pdb6SUC/pdb] (PTPRU-D1 oxidised form). The source data underlying Figs. 1c–e, g–j, 2f, 3b–e, 4c, 5b–g and Supplementary Figs. 3a, b, 4a, b, d, 7, 8a–c, and 9a, b are provided as a source data file. Other data and reagents are available from the corresponding authors upon reasonable request.

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

## Acknowledgements

We acknowledge Diamond Light Source for time on beamlines I03 and I04 under proposal MX15916. Remote access was supported in part by the EU FP7 infrastructure grant BIOSTRUCT-X (Contract No. 283570). We thank Stephen Graham for assistance with the crystallography work and the CIMR flow cytometry core facility for their assistance in cell-sorting. I.M.H. is supported by a CIMR PhD studentship. M.K. acknowledges funding from the German Research Foundation (DFG) under Germany's Excellence Strategy (CIBSS – EXC-2189 – Project ID 390939984 and BIOSS – EXC-294). H.J.S. and G.W.F. are funded by a Sir Henry Dale Fellowship jointly funded by the Wellcome Trust and the Royal Society (109407/Z/15/Z). J.E.D. is funded by a Royal Society University Research Fellowship (UF150682).

## Author contributions

I.M.H., H.J.S., and J.E.D. conceived and designed the experiments. I.M.H., G.W.F., P.R., and M.K. performed biochemical experiments. I.M.H. and J.E.D. performed structural refinement and analysis. I.M.H., H.J.S., and J.E.D. wrote the manuscript.

## Competing interests

The authors declare no competing interests.
