## [Peer Review File · Nature Communications]

Reviewers' comments:

Reviewer #1 (Remarks to the Author):

NCOMMS-19-37816

The manuscript by Hay et al describes the structure and characterization of the PTPRU pseudophosphatase, a receptor tyrosine phosphatase that is proposed to be proto-oncogenic, although its true function is still not fully established. The biological function of this protein is the question addressed in this manuscript, which describes biochemical, structural and cell biological data unified by an effort to uncover its true function in cells. The experiments are solid and interesting, providing new intriguing possibilities about the role of this protein in biology and cancer. That being said, some of the conclusions lack the solidity one might expect from a Nature Communications paper and thus a few additional studies are proposed that are designed to strengthen the manuscript conclusions, especially regarding mechanism. Unraveling these questions are very important to the field of phosphorylation signaling.

1.Activity assays: The enzyme activity assays are solid and convincing. However, no rationale is provided as to why PTPRK was selected for the comparative studies and a sentence or two at the beginning of this paragraph would strengthen this entire section (it took a couple of looks at the figure to realize some of the data refer to PTPRK). If the PTP loop Cys is replaced by Ser, are there any phenotypes? Similarly, is it perfectly conserved across all organisms that contain PTPRU genes? This might also provide insights into the role of this protein (and confirm lack of activity is 'fine').

2.Structure: This section is very interesting, but a few additional experiments would strengthen the conclusions. The loss of an organized pTYR substrate specificity loop could certainly impact activity [although, this loop is often expected to restrict substrates to pTYR and one *might* predict that its absence could render the protein more active and certainly to more substrates, ie pS or pT]. Further, the authors discover an unusual conformation for the PTP loop originally hypothesized (but later proven by the authors to be the sole cause of PTPRU inactivity) that this is due to the presence of a Thr residue when its normally Ala, whose conformation also restricts pY access. To solidify the conclusions, the following experiments are suggested:

a.CHIMERA: Make a chimera to test the pTyr substrate recognition loop hypothesis: replace the PTPRU residues 904-925 with the corresponding residues from PTPRK and visa versa. Test activity. Additional structures (if possible) would make this section stronger but crystallography is never predictable and thus not required. But activity assays are absolutely necessary. For the chimera's, also perform the proteolytic cleavage and stability assays. This should provide strong confirmation if the author hypothesis is true.

b. If the PTPRU-PTPRK904-925 (these exact residue numbers are likely different as these are the PTPRU numbers) chimera is not active, add the T->A mutation in the PTP loop as well (and, visa versa for the PTPRK-PTPRU(904-925)). Test activity.

c. Together, these experiments will provide the data needed to conclude that these two sequence/structural differences are what renders PTPRU inactive.

RMSD: 1.014—too many significant digits (for the accuracy of crystal structures); 1.0 will suffice.

The crystallization pH is unusually high for PTPs. If another crystal form appeared, one might collect to determine if the structure is the same across different pH values.

See also the comment in the figure—namely that from the figure 3A there *seems* to be a second PTPRU loop that may be the cause of the unusual PTP loop conformation. If this second loop does, in fact, overlap with the 'typical' position for the residue that corresponds to the Thr of the PTP loop, additional experiments to test this are warranted.

3. Oxidation. While interesting, the biological importance is unclear (since it is not an active phosphatase). Based on the data presented, one might ask, would abolishing the ability to form a backdoor disulfide impact PTPRU function? i.e., if the backdoor cys was mutated to ser, would there be any phenotypes?

Figure edits.

Figure 1:

A. This would benefit with something that 'highlights' the members of this family (M/K/T/U)—is there anything different about them that one might be able to add to this figure? Maybe, difference in activity; something else? Why are the D domains different colors—does this have an implied meaning (if it does, it is unclear what it is; green/red are mentally correlated with 'go'/'stop' and one might pick different colors if this is not what is implied).

B. Since the manuscript (and experiments) are focused on 'U' and 'K', put these at the top of the alignments, not the bottom, with 'U' at the top.

B-G. Exploit color. Use a different color for the font for PTPRK and PTPRU text (on gels, kinetic assays, etc) for all labels so its easier for readers to 'distinguish' the data corresponding to different constructs—these names are too similar for the non-experts and it takes a few looks before one realizes the constructs differ—make it obvious.

Figure 2:

A and B do not really illustrate what the authors are trying to show. For example, it is impossible to 'see' where the disordered loop 'ends' are in PTPRU (they are not even labeled). Instead, show an example of how the pY SRL restricts access to only pY residues (using one of the PTP structures with a pY at the active site). Only real experts understand the importance of this loop without a figure-- give the broad audience of NC the chance to understand the argument.

With the overlay in A, also show each structure separately (this will make the 'missing' loop more obvious). In the overlay, also make the pY loop (missing residues) light grey in the PTPRK structure; visually, this will be easier for the reader to 'see' what is missing.

B is much less useful and might be fine for supplemental. Alternatively, just the overlaid pY loops as an 'inset' somewhere will illustrate the point the authors are making. Also, for this overlay, one might try ribbon instead of cartoon in pymol (some of the secondary structural elements are not uniformly defined among all structures—they likely overlay really perfectly with ribbon).

C. put a triangle indicating increasing concentrations of trypsin over the numbers—visually easier. [Some of this figure may change based on the recommended chimera experiments.]

Figure 3.

A. State at least the number of PDBs overlaid in the legend. Label T1089. Also, it is clear from ii and iii in this panel that there is another PTPRU loop that 'appears' to overlap the residues that correspond to the 'T' in PTPRU (looks like alanines and some other amino acids)—that is, is this OTHER loop the reason the PTP loop moves? i.e., because there is a second loop in PTPRU to pushes is up out of the way. It is a little hard to see here but this might explain why the PTP loop is in such an unusual conformation (if it actually does clash with the 'canonical position for the Thr residue sidechain).

C. This figure is not useful as it appears that it is trying to make the same point as panel iii in 'A'. Delete it.

Figure 4: move to supplemental? Its simply unclear if this ability of PTPRU has any biological relevance and thus it may be more artifactual (not lost in evolution) than anything else.

Fig 5, A. The different colors are very helpful. Indicating which domains are active on this figure would be extremely helpful for more easily interpreting the rest of this figure.

English: Reread for English. There are a handful of typos and awkward sentences.

Reviewer #2 (Remarks to the Author):

The receptor PTPRU is a redox sensitive pseudophosphatase

Hay Deane

NCOMMS-19-37816

The authors present a detailed biochemical analysis of PTPRU. Using a variety of approaches, they have not been able to demonstrate phosphatase activity in their preparations of PTPRU. On this basis, they conclude that PTPRU is catalytically inactive. They present a crystal structure of the membrane-proximal, D1, domain of PTPRU and note some structural anomalies compared to known catalytically-competent PTP domains. They note oxidation of the "catalytic" Cys residue in PTPRU D1 and engagement of this Cys by a "back-door" Cys residue in a disulfide bond, as has been reported for other PTPs. They also present domain swapping studies and suggest that the data should be interpreted by a mechanism of action in which "PTPRU can bind substrates and protect them from dephosphorylation by related phosphatases". Overall, the studies are thorough and well-performed. Nevertheless, I feel like this is Bertrand Russell's Teapot territory. The authors are trying to prove a negative – that a protein that looks like a member of the PTP family does not possess intrinsic catalytic activity. This is a difficult task but, as with the teapot, I think the burden of proof lies with

them to demonstrate conclusively that PTPRU is not a phosphatase. In this case, I do not think they have done so satisfactorily. What is left is a series of biochemical observations that are not unique and have been reported already for other members of the PTP family; as such, I do not think this rises to the level required for Nature Communications.

I think the pseudoenzyme field is a fascinating area of research. Unfortunately, it is replete with examples of proteins that were classified originally as inactive pseudoenzymes, but with further analysis were shown actually to be enzymatically active. The kinases provide a number of such examples – including HER3, KSR, JH2 domain in JAK etc. The protein tyrosine phosphatases (PTPs) themselves include several such cases. It is interesting to note that the authors, in their Introduction on page 3, highlighted PTPN23 (HD-PTP), PTPN14 (PTPD2) and PTPN21 (PTPD1) as being catalytically inactive. Nevertheless, there are papers published, by several different labs, attesting to the fact that these PTPs do possess intrinsic activity. The authors should have presented a more thorough and balanced introduction. Furthermore, this suggests that great caution should be exercised before concluding from limited biochemical analysis that a given protein is a pseudoenzyme – it may just be that you have yet to test the correct substrate.

The authors place emphasis on the observation that several important sequence motifs in PTPRU have diverged from the canonical motifs observed in catalytically competent PTPs. In particular, “PTPRU possesses non-canonical WPD (WPE) and pTyr recognition loop sequences (GSRQ rather than KNRY), as well as a unique threonine in the PTP loop”. Nevertheless, it is important to note that other PTPs, such as PTPRQ and PTPN23, also display a WPE motif, but still possess intrinsic activity. Even in the study of PTP1B cited by the authors, it was reported that although mutation of the D in the WPD loop was inhibitory, the mutant protein maintained a measurable level of activity. This raises the question, just how much activity against an artificial substrate *in vitro* is necessary to underlie a physiological function? It should be noted that like PTPRU, PTPN23 also presents a non-canonical pTyr recognition loop. Finally, the authors themselves report here that “introduction of a threonine to the active site was not sufficient to inactivate PTPRK D1” (see page 6) and, therefore, this Thr in PTPRU is unlikely to be the cause of their failure to demonstrate activity. Consequently, I think one should be cautious about placing too much reliance on such sequence analyses.

The data presented in this report suggest that PTPRU was a difficult PTP to work with. For example, on page 7, the authors state that “we were unable to purify full PTPRU-ICD (D1 +D2)” and they were unable to generate chimeras containing PTPRU D2. Most of their analysis of potential activity in PTPRU was conducted on a truncated construct expressing the isolated D1 domain as a recombinant protein in *E. coli*. How do they know that the protein folded properly? There have been reports to suggest a chaperone function for D2 domains – perhaps the presence of D2 is essential for manifestation of catalytic activity in PTPRU? Also, it was reported by Philip Cohen’s lab that, at least for protein phosphatase 1, the activity of the protein expressed in *E. coli* possessed different features from the activity of the phosphatase purified from mammalian tissues [MacKintosh et al

(1996) FEBS Letts 397: 235-238]. With this in mind, the presence of disordered segments in the structure of the recombinant protein may reflect incorrect folding in *E. coli*.

Reviewer #3 (Remarks to the Author):

In the manuscript, “The receptor PTPRU is a redox sensitive pseudophosphatase,” Hay et al provide evidence that the protein tyrosine phosphatase D1 domain (PTP D1) of the receptor phosphatase PTPRU lacks catalytic activity. The authors also show that the protein’s D1 domain is less thermostable than a homologous PTP domain and they solve the first crystal structure of PTPRU’s PTP D1 domain, uncovering deviations from canonical PTP-domain structure that are consistent with PTPRU D1’s lack of activity and instability. Finally, the authors provide preliminary evidence that that the intracellular portion of PTPRU can associate with substrates of other receptor PTPs (e.g., p120-catenin), sequestering them from potential dephosphorylation; when PTPRU is knocked out, the tyrosine phosphorylation of p120-catenin counterintuitively decreases.

Receptor PTPs are key regulators of cell signaling, and the demonstration that a member of the family is not really a PTP (i.e., a pseudophosphatase) would represent a significant finding. The manuscript is well written and the experiments are well designed and clearly presented. However, a revised version of the manuscript would need to address the following critiques before I could recommend it for publication.

Critique 1: Cropping of anti-phosphotyrosine westerns (Figures 1E, 1G, and 5E). The anti-pTyr blots purport to demonstrate the lack of activity of the PTPRU D1 against a whole range of potential substrates. So, why are the blots cropped at seemingly arbitrary molecular weight cut-offs? The authors should present the uncropped blots to show the tyrosine-phosphorylation levels over the entire molecular-weight range of proteins in the treated lysates.

Critique 2: How have the authors controlled for the possibility that the lack of activity in their PTPRU D1 constructs may be an artifact of truncating the protein? The thermal instability of the protein—as well as the mysterious inability to express the D1+D2 construct—suggest that this protein doesn’t “like” to be truncated. The authors should attempt experiments in which they assess activity in the context of the full-length protein. Perhaps immunoprecipitating PTPRU side-by-side with another RTP and then comparing the PTP activities in the IPs?

Critique 3: The authors should more fully engage the previous literature that suggests that PTPRU is an active phosphatase. The authors state only in passing in the introduction that "... previous work suggests that PTPRU can dephosphorylate substrates such as β -catenin." Indeed, one of the papers that the manuscript references (Yan et al, 2006) shows rather convincing evidence that expression of PTPRU causes a dramatic decrease in phosphorylation of β -catenin's Y102, and that this decrease is dependent on the presence of catalytically active PTPRU (the dephosphorylation is completely lost when an inactive Cys to Ser mutant is expressed instead of the wild-type protein). How do the authors square this previous finding with the lack of PTP activity in their constructs? Have they attempted to look at PTPRU's activity with this previously reported substrate?

Response to Reviewers' comments:

Reviewer #1 (Remarks to the Author):

NCOMMS-19-37816

The manuscript by Hay et al describes the structure and characterization of the PTPRU pseudophosphatase, a receptor tyrosine phosphatase that is proposed to be proto-oncogenic, although its true function is still not fully established. The biological function of this protein is the question addressed in this manuscript, which describes biochemical, structural and cell biological data unified by an effort to uncover its true function in cells. The experiments are solid and interesting, providing new intriguing possibilities about the role of this protein in biology and cancer. That being said, some of the conclusions lack the solidity one might expect from a Nature Communications paper and thus a few additional studies are proposed that are designed to strengthen the manuscript conclusions, especially regarding mechanism. Unraveling these questions are very important to the field of phosphorylation signaling.

> We thank the reviewer for their thorough and constructive comments, which we have addressed below.

1. Activity assays: The enzyme activity assays are solid and convincing. However, no rationale is provided as to why PTPRK was selected for the comparative studies and a sentence or two at the beginning of this paragraph would strengthen this entire section (it took a couple of looks at the figure to realize some of the data refer to PTPRK). If the PTP loop Cys is replaced by Ser, are there any phenotypes? Similarly, is it perfectly conserved across all organisms that contain PTPRU genes? This might also provide insights into the role of this protein (and confirm lack of activity is 'fine').

> We have made several changes to the text and figures to address this very constructive comment. In the second sentence of the results section we now explain the relevance of using PTPRK, as the closest paralog to PTPRU, and have moved our similarity/identity analysis of this receptor family to Supplementary Table 1. We also now include an alignment of PTPRU from several vertebrate species (Supplementary Fig. 2) to highlight the complete conservation of the catalytic and backdoor cysteines, as well as alterations in other key PTP motifs. The complete conservation of the catalytic cysteine in PTPRU across species despite the lack of phosphatase activity suggests it may play an alternative role as now detailed below in our response to the subsequent comment regarding oxidation. All of our activity assays are carried out in the presence of reducing agents (DTT or TCEP) to avoid the potential of disulphide bond formation interfering with these assays. We have now replaced the PTP loop cysteine with Ser and used this construct in new IP-activity assays of the intact intracellular domain (new Fig. 1f-h) demonstrating it has no effect on PTPRU phosphatase activity as expected. However, experiments to test changes in "phenotype" such as differences in binding PTPRK substrates as demonstrated in Fig. 5 will constitute part of extensive follow-up work.

*2. Structure: This section is very interesting, but a few additional experiments would strengthen the conclusions. The loss of an organized pTYR substrate specificity loop could certainly impact activity [although, this loop is often expected to restrict substrates to pTYR and one *might* predict that its absence could render the protein more active and certainly to more substrates, ie pS or pT]. Further, the authors discover an unusual conformation for the PTP loop originally hypothesized (but later proven by the authors to be the sole cause of PTPRU inactivity) that this is due to the presence of a Thr residue when its normally Ala, whose conformation also restricts pY access.*

> The reviewer's statement here that we showed that the unusual Thr in the PTP loop is the sole cause of PTPRU-D1 activity is incorrect and so we have endeavoured to clarify this as follows. Our previous data (old Figure 3b) showed that the Thr in PTPRU is not solely responsible for its inactivity as the equivalent mutation in PTPRK retains activity. Also, mutation of the Thr alone to an Ala does not restore activity to PTPRU showing that there are additional/other contributing factors to its lack of activity. To clarify this point, we have repeated these experiments and combined them with new data (described below) in an additional results section addressing the role of multiple motifs in PTPRU on activity and stability. Finally, in the original manuscript we alluded to an analysis of pSer and pThr. We now show these data in Supplementary Fig 4 and state more clearly that PTPRU does not appear to be a Ser/Thr phosphatase.

To solidify the conclusions, the following experiments are suggested:

a. CHIMERA: Make a chimera to test the pTyr substrate recognition loop hypothesis: replace the PTPRU residues 904-925 with the corresponding residues from PTPRK and visa versa. Test activity. Additional structures (if possible) would make this section stronger but crystallography is never predictable and thus not required. But activity assays are absolutely necessary. For the chimera's, also perform the proteolytic cleavage and stability assays. This should provide strong confirmation if the author hypothesis is true.

> We have now made chimeric proteins where the pTyr loops of PTPRK and PTPRU have been swapped in order to test the impact of this loop on protein activity and stability. To avoid confusion with the naming of the D1/D2 swapped chimeras used later in the manuscript we have called these new proteins PTPRK-D1 U-pTyr and PTPRU-D1 K-pTyr and provide a schematic and sequence alignment for clarity (new Figure 3a). These proteins were subjected to activity assays and limited proteolysis (new Figure 3b-e). We have also carried out the limited proteolysis using both subtilisin and trypsin to avoid any sequence-specific effects of these two enzymes. The trypsin data are included as supplementary data (Supplementary Fig. 7). In our new data, we show that the PTPRK pTyr loop, either alone or in combination with the E1053D mutation, does not restore activity to PTPRU (new Fig. 3b) and indeed the K loop appears to further destabilise the PTPRU-D1 (new Fig. 3d). Introduction of the PTPRU loop into PTPRK is sufficient to render PTPRK inactive (new Fig. 3b). This inactivity is likely the result of the inability of the PTPRU loop to adopt an ordered conformation as this chimera is significantly less resistant to proteolysis than the wild-type PTPRK (new Fig. 3e). Therefore, although a single change to PTPRK (swapping the pTyr loop) can inactivate it, not even multiple changes to PTPRU can restore activity.

By making these data part of a new distinct results section and an additional figure we hope it will be clearer to readers what the contribution of each change is to the activity and stability of these enzymes.

Although the thermal stability data for wild-type PTPRU and PTPRK identifies that PTPRU has a lower overall melting temperature than PTPRK (now Fig. 2f) we were unable prior to shutdown of our laboratories in the Covid19 crisis to fully test this effect on all the mutant and chimeric forms of PTPRU and PTPRK.

b. If the PTPRU-PTPRK904-925 (these exact residue numbers are likely different as these are the PTPRU numbers) chimera is not active, add the T->A mutation in the PTP loop as well (and, visa versa for the PTPRK-PTPRU(904-925)). Test activity.

> As described above and shown in Fig. 3c it is clear from our data that the Thr in the PTP loop is not the primary driver of inactivity as the introduction of a Thr into PTPRK does not render it inactive (Fig. 3c).

c. Together, these experiments will provide the data needed to conclude that these two sequence/structural differences are what renders PTPRU inactive.

RMSD: 1.014—too many significant digits (for the accuracy of crystal structures); 1.0 will suffice.

> These have now been updated as suggested (Supplementary Table 2).

The crystallization pH is unusually high for PTPs. If another crystal form appeared, one might collect to determine if the structure is the same across different pH values.

> Although pH 9 is higher than the pH at which the majority of PTP structures have been determined, there are several determined at pH 8.5 (PDB 2NZ6, 2A8B and 2A3K) and one at pH 11 (PDB 1WCH) which all possess high structural similarity to PTPRU (RMSD 1.0-1.2 Å, Supplementary Table 2) but retain well-ordered pTyr and PTP loops. This suggests that the disorder of the pTyr loop and rearrangement of the PTP loop in PTPRU-D1 is unlikely to be driven by the pH of the crystallisation condition. Unfortunately we do not currently have a second crystal form of PTPRU in order to test this further experimentally.

See also the comment in the figure—namely that from the figure 3A there *seems* to be a second PTPRU loop that may be the cause of the unusual PTP loop conformation. If this second loop does, in fact, overlap with the 'typical' position for the residue that corresponds to the Thr of the PTP loop, additional experiments to test this are warranted.

> The reviewer is correct that the loop visible in the original Fig. 3a is partially overlapping with the 'typical' position for the residues that correspond to T1089 of the PTP loop. This loop (C1121-M1127) is mentioned in the text (for both the reduced and oxidised structures) but we acknowledge that it was not clear how the positions of the PTP and C1121-M1127 loops are positioned relative to each other. We have now included an extra supplementary figure (Supp. Fig. 6) and rewritten the text to clarify these points. Although it is hard to imply causality when it comes to potentially inter-dependent conformational changes we do not think that the orientation of the PTP loop is caused by the C1121-M1127 loop for two reasons now described in the text but also summarised here. Firstly, the T1089 sidechain is stabilised in the new conformation via a hydrogen bond to the R1091 backbone capping the nearby helix, this hydrogen bond cannot be formed by the hydrophobic sidechains present in other classical PTPs. Therefore, this hydrogen bond may be sufficient to orient the PTP loop in this new conformation. Secondly, the C1121-M1127 loop is not well ordered in the reduced form (although it was possible to build a single conformation the density is not as convincing as nearby regions) and this loop is disordered in the oxidised form (there was no electron density in which to build the loop). This suggests that the C1121-M1127 loop in PTPRU may adopt multiple conformations. This mobility, combined with the stabilising hydrogen bond formed between T1089 and R1091, suggests that the PTP loop conformation is unlikely to be caused by the C1121-M1127 loop. Possibly the best way to experimentally test this would be a structure of PTPRU possessing the C1121R mutation. However, as the reviewer noted earlier "crystallography is never predictable" and sadly, shutdown of our laboratories precludes any attempts at this.

3. Oxidation. While interesting, the biological importance is unclear (since it is not an active phosphatase). Based on the data presented, one might ask, would abolishing the ability to form a backdoor disulfide impact PTPRU function? i.e., if the backdoor cys was mutated to ser, would there be any phenotypes?

> The low pKa of PTP catalytic cysteines enables them to act as nucleophiles, but also renders them sensitive to oxidation. The chemical reactivity of such cysteine residues compared to other amino acids means that they are less likely to be randomly conserved in proteins (e.g. Marino & Gladyshev. 2010). We think it is potentially interesting that this PTP has preserved redox sensing, despite losing catalytic activity in evolution. As the reviewer highlights, the current model for oxidation-mediated regulation of PTPs is through their reversible inactivation. What if these are in fact two distinct functions? For example, several pseudokinases preserve ATP binding that is key to their function. Perhaps redox sensing is a preserved function for pseudophosphatases. Additional examples can be found amongst the catalytically inactive D2 domains of other RPTPs that indicate a similar preservation of redox-sensing. In our discussion we mention PTPRA oxidation, which can induce conformational changes (Groen et al 2005), mediate intermolecular disulphides (Van der Wijk et al. 2004) and promote rotational coupling (Van der Wijk et al. 2003). Finally, a recent Nat Chem Biol paper, from the Tonks and Boivin labs, highlights the importance of a redox-mediated conformational change affecting the PTP1B pTyr loop and its ability to interact with 14-3-3 proteins. We also note that PTPRU oxidation was previously observed in a global study of PTP oxidation in cells (Karisch et al. 2011). We therefore think that this warrants further research and should stay as a main feature of the manuscript.

Figure edits.

Figure 1:

A. This would benefit with something that ‘highlights’ the members of this family (M/K/T/U)—is there anything different about them that one might be able to add to this figure? Maybe, difference in activity; something else? Why are the D domains different colors—does this have an implied meaning (if it does, it is unclear what it is; green/red are mentally correlated with ‘go’/‘stop’ and one might pick different colors if this is not what is implied).

> Figure one has been substantially modified in terms of its colouring and labelling to address this point and those below.

B. Since the manuscript (and experiments) are focused on ‘U’ and ‘K’, put these at the top of the alignments, not the bottom, with ‘U’ at the top.

> Thank you for this suggestion. All alignments now focus on PTPRU and PTPRK (e.g. Fig 1b).

B-G. Exploit color. Use a different color for the font for PTPRK and PTPRU text (on gels, kinetic assays, etc) for all labels so its easier for readers to ‘distinguish’ the data corresponding to different constructs—these names are too similar for the non-experts and it takes a few looks before one realizes the constructs differ—make it obvious.

> We have now exploited the use of colour to make it more visually intuitive whether the data are related to PTPRK and PTPRU.

Figure 2:

A and B do not really illustrate what the authors are trying to show. For example, it is impossible to ‘see’ where the disordered loop ‘ends’ are in PTPRU (they are not even labeled).

> The new Fig. 2a clearly illustrates where the disordered loop begins and ends.

Instead, show an example of how the pY SRL restricts access to only pY residues (using one of the PTP structures with a pY at the active site). Only real experts understand the

importance of this loop without a figure-- give the broad audience of NC the chance to understand the argument.

> We have now included in Fig. 2b and 2c clear illustrations of how the pTyr loop is normally folded and how this defines the substrate binding pocket specificity for pTyr.

With the overlay in A, also show each structure separately (this will make the 'missing' loop more obvious). In the overlay, also make the pY loop (missing residues) light grey in the PTPRK structure; visually, this will be easier for the reader to 'see' what is missing.

> The new Fig. 2 panels a and b based on the comments above now address this request.

B is much less useful and might be fine for supplemental. Alternatively, just the overlaid pY loops as an 'inset' somewhere will illustrate the point the authors are making. Also, for this overlay, one might try ribbon instead of cartoon in pymol (some of the secondary structural elements are not uniformly defined among all structures—they likely overlay really perfectly with ribbon).

> Panel b has been removed and the new Fig. 2b based on comments above also includes an inset illustrating the overlaid pTyr loops. We also thank the reviewer for the suggestion of using ribbon in PyMol and agree this is much clearer.

C. put a triangle indicating increasing concentrations of trypsin over the numbers—visually easier. [Some of this figure may change based on the recommended chimera experiments.]

> The equivalent of panel c is now part of Fig. 3 and includes the chimera experiments as recommended. A triangle representing increasing protease concentrations is now included for clarity in Figures 3c-e and Supp. Fig. 7.

Figure 3.

A. State at least the number of PDBs overlaid in the legend. Label T1089.

> The number of overlaid PDBs used (in new Fig. 2b and 2d) is now included in the legend and the identity of these structures is also included in Supplementary Table 2. T1089 is also labelled.

Also, it is clear from ii and iii in this panel that there is another PTPRU loop that 'appears' to overlap the residues that correspond to the 'T' in PTPRU (looks like alanines and some other amino acids)—that is, is this OTHER loop the reason the PTP loop moves? i.e., because there is a second loop in PTPRU to pushes is up out of the way. It is a little hard to see here but this might explain why the PTP loop is in such an unusual conformation (if it actually does clash with the 'canonical position for the Thr residue sidechain).

> This comment has been addressed above in the related comment resulting in substantial additional commentary in the main text and an addition of Supplementary Fig. 6.

C. This figure is not useful as it appears that it is trying to make the same point as panel iii in 'A'. Delete it.

> A modified version of this panel is now part of Fig. 2 and addresses a request above for a clearer explanation of how pTyr binding depends on the folded pTyr loop. We think this new panel 2e, when compared with 2c, is extremely useful for illustrating to a general audience how the structural rearrangements in PTPRU would hinder pTyr binding.

Figure 4: move to supplemental? Its simply unclear if this ability of PTPRU has any biological relevance and thus it may be more artifactual (not lost in evolution) than anything else.

> Please see our detailed answer to point 3 above regarding our reasons for why oxidation may be biologically relevant and why we feel it is important to retain this observation and figure in the main text.

Fig 5, A. The different colors are very helpful. Indicating which domains are active on this figure would be extremely helpful for more easily interpreting the rest of this figure.

> The schematic in Fig. 5a has been updated to include a representation of active versus inactive domains.

English: Reread for English. There are a handful of typos and awkward sentences.

> We have made additions to the text and corrected any English and awkward sentences.

Reviewer #2 (Remarks to the Author):

The receptor PTPRU is a redox sensitive pseudophosphatase

Hay Deane

NCOMMS-19-37816

The authors present a detailed biochemical analysis of PTPRU. Using a variety of approaches, they have not been able to demonstrate phosphatase activity in their preparations of PTPRU. On this basis, they conclude that PTPRU is catalytically inactive. They present a crystal structure of the membrane-proximal, D1, domain of PTPRU and note some structural anomalies compared to known catalytically-competent PTP domains. They note oxidation of the “catalytic” Cys residue in PTPRU D1 and engagement of this Cys by a “back-door” Cys residue in a disulfide bond, as has been reported for other PTPs. They also present domain swapping studies and suggest that the data should be interpreted by a mechanism of action in which “PTPRU can bind substrates and protect them from dephosphorylation by related phosphatases”. Overall, the studies are thorough and well-performed.

> We are pleased that the reviewer found the studies to be thorough and well executed.

Nevertheless, I feel like this is Bertrand Russell’s Teapot territory. The authors are trying to prove a negative – that a protein that looks like a member of the PTP family does not possess intrinsic catalytic activity. This is a difficult task but, as with the teapot, I think the burden of proof lies with them to demonstrate conclusively that PTPRU is not a phosphatase. In this case, I do not think they have done so satisfactorily. What is left is a series of biochemical observations that are not unique and have been reported already for other members of the PTP family; as such, I do not think this rises to the level required for Nature Communications.

> While the authors agree with the reviewer that proving a negative is a difficult task, we do not agree that we are in Bertrand Russell’s Teapot territory. The celestial teapot is an analogy for a situation in which no scientific evidence exists. Moreover, we set out to test for

activity rather than prove its absence. Another way to frame our findings is that we find no evidence for PTPRU catalytic activity. It should be noted that one previous study did show activity of purified PTPRU domains in a pNPP assay (Crossland et al. 1996), however, no mutant control was used and the D2 domain showed equal activity to the D1. Of note, the PTPRU D2 domain lacks key catalytic residues of the WPD (WSA), KNRY (KNRS) and Q loops (ExxxQ). However, we do agree that it is our responsibility to provide evidence to support our claims, as for any study.

We have provided substantial evidence using robust and well-executed scientific methods to test whether PTPRU is an active phosphatase. We applied a range of different biochemical approaches and phosphatase assays demonstrating that bacterially or mammalian-expressed PTPRU-D1, in the presence or absence of the D2 domain, does not display any activity against a variety of substrates. We used X-ray crystallography, limited proteolysis and thermal denaturation studies to demonstrate that despite no detectable phosphatase activity, PTPRU is a folded protein that resembles other PTPs but with distinct features that go toward explaining its lack of activity. We go on to assess the individual contributions of sequence variants, concluding that no one feature can explain inactivity alone. We then use cell biology approaches to demonstrate that despite its lack of identifiable enzyme activity, PTPRU plays a role in specific protein interactions. The evidence collected very strongly indicates that PTPRU is intrinsically catalytically inactive but functionally active in cells.

I think the pseudoenzyme field is a fascinating area of research. Unfortunately, it is replete with examples of proteins that were classified originally as inactive pseudoenzymes, but with further analysis were shown actually to be enzymatically active. The kinases provide a number of such examples – including HER3, KSR, JH2 domain in JAK etc.

> We agree that this is a fascinating field, where the classical enzyme rules are broken. Indeed many bona fide pseudoenzymes conserve some functions associated with catalysis meaning the overall protein architecture largely resembles the catalytically competent counterparts. For example, KSR, HER3 and TRIB pseudokinases still bind to ATP, and limited levels of phosphotransfer can be detected. However, classical catalysis is significantly impaired and the importance of the residual activity is not always well established. Furthermore, these examples from the kinase field are still widely referred to as pseudokinases. The best example of inactivity being wrongly assigned is the CASK kinase, which was resolved using an X-ray crystal structure. Thus, the combination of robust biochemical assays, X-ray crystallography and cell based experiments represents a gold standard in studying such pseudoenzymes. It remains a possibility that PTPRU is competent to perform alternative functions, beyond the removal of phosphate. If this were the case, we would still label PTPRU a pseudophosphatase.

The protein tyrosine phosphatases (PTPs) themselves include several such cases. It is interesting to note that the authors, in their Introduction on page 3, highlighted PTPN23 (HD-PTP), PTPN14 (PTPD2) and PTPN21 (PTPD1) as being catalytically inactive. Nevertheless, there are papers published, by several different labs, attesting to the fact that these PTPs do possess intrinsic activity. The authors should have presented a more thorough and balanced introduction. Furthermore, this suggests that great caution should be exercised before concluding from limited biochemical analysis that a given protein is a pseudoenzyme – it may just be that you have yet to test the correct substrate.

> We have altered the introduction to be more balanced with respect to the literature on these putative pseudoPTPs. However, caution is also required in interpreting “positive” evidence of activity. For example, for PTPN23 there are two studies using purified domains suggesting it is catalytically active, and two studies indicating inactivity. Mariotti et al. express wildtype PTPN23 in HEK293 cells, immunoprecipitate it and perform a DiFMUP

assay (similar to revision experiments we have performed - see below). PTPN23 activity was detectable, which could be partially inhibited by the PTP inhibitor vanadate. A major caveat of this study was the absence of a cysteine mutant control. These data were later refuted by a better controlled study that demonstrated the previously observed activity was most likely due to contaminating phosphatase activity. The authors of the more recent study went on to show that restoration of a key residue in the PTP loop resulted in detectable PTPN23 activity (Gingras et al. 2009). Similar to our results with PTPRK, PTPN23 phosphatase activity was observed in the presence of a WPE, rather than WPD, motif.

A second study (Zhang et al. 2017) reports specific dephosphorylation of Fyn kinase by bacterially expressed PTPN23, including a number of appropriate controls. This is in contrast to Barr et al. who could not identify any activity against a panel of phosphopeptides or DiFMUP. Taken together, these results suggest that PTPN23 has very restricted substrate specificity, and perhaps even an induced fit model could be evoked, which is nevertheless unusual when compared to other PTPs. There is currently no crystal structure of PTPN23, and this might be required to resolve these discrepancies.

Two biochemical studies suggest that PTPN14 and PTPN21 are inactive.

1. Barr et al. show limited activity in a DiFMUP assay and no activity against a panel of phosphopeptides.
2. Substitution of an isoleucine (found in the PTPN14 pTyr recognition loop) into the closely related PTPN3 was shown to be inactivating (Chen et al. 2015).
3. Substitution of WPE (as found in PTPN21) into PTPN3 caused a 380 fold decrease in kCat, with no activity detected in equivalent assays for either PTPN14 or PTPN21. (Chen et al 2015).

On the contrary, there are several reports of specific PTPN14 and PTPN21 substrates. It is possible that, like for PTPN23, specific protein substrates exist, however, similar caution should be exercised. For example, a cysteine mutant should be used to demonstrate that purified protein activity is not a consequence of contaminating phosphatases (e.g. missing in Choi et al (2018)). Additionally, hyperphosphorylation of a protein in KO cells does not mean it is a direct substrate (e.g. Ni et al. (2019)).

For PTPRU, we have used both mammalian and bacterially-expressed constructs to test activity. We demonstrate that substitution of PTPRU sequences into the active paralog PTPRK significantly impairs its activity, and provide a high resolution crystal structure that indicates an obstructed active site. While we agree these data do not absolutely prove the absence of hydrolase activity, the data strongly indicate that PTPRU is atypical amongst PTPs and using standard phosphatase assays we detect no catalytic activity.

The authors place emphasis on the observation that several important sequence motifs in PTPRU have diverged from the canonical motifs observed in catalytically competent PTPs. In particular, “PTPRU possesses non-canonical WPD (WPE) and pTyr recognition loop sequences (GSRQ rather than KNRY), as well as a unique threonine in the PTP loop”. Nevertheless, it is important to note that other PTPs, such as PTPRQ and PTPN23, also display a WPE motif, but still possess intrinsic activity. Even in the study of PTP1B cited by the authors, it was reported that although mutation of the D in the WPD loop was inhibitory, the mutant protein maintained a measurable level of activity. This raises the question, just how much activity against an artificial substrate in vitro is necessary to underlie a physiological function?

> We agree that there are many examples of PTPs with either naturally occurring or engineered “WPE motifs” that display detectable catalytic activity. We find the same is true for PTPRK, which possesses detectable activity upon introduction of “WPE” (Fig 1i), albeit

>100-fold lower than wildtype. We likewise conclude that this change is insufficient to explain the absence of PTPRU activity. We have now introduced an entire section dedicated to the dissection of the roles of each motif variant in PTPRU, concluding that no individual motif variant explains inactivity.

It should be noted that like PTPRU, PTPN23 also presents a non-canonical pTyr recognition loop. Finally, the authors themselves report here that “introduction of a threonine to the active site was not sufficient to inactivate PTPRK D1” (see page 6) and, therefore, this Thr in PTPRU is unlikely to be the cause of their failure to demonstrate activity. Consequently, I think one should be cautious about placing too much reliance on such sequence analyses.

> As mentioned above, in canonical phosphatase assays PTPN23 has been reported to be inactive and could be reactivated by mutating the PTP loop. Thus, the WPE and KNRH motifs within PTPN23 are not responsible for the inactivity reported. KNRH is much closer to the canonical KNRY, than GSRQ (found in PTPRU-D1). As the imidazole ring of histidine is also aromatic, it is possible that this residue can still pack against phosphotyrosine substrates. Furthermore, we now show that replacing the pTyr loop of PTPRK with that of PTPRU is sufficient to completely inactivate it (Fig 3b). As stated above and in the modified text, we have clarified that we do not think that any single sequence variation in PTPRU is responsible for its inactivity, rather it is a combination of multiple changes that together contribute to its inactivity. As shown in the new Fig. 3 neither T1089, E1053D or swapping the pTyr loops or combinations of these restore activity to PTPRU.

The data presented in this report suggest that PTPRU was a difficult PTP to work with. For example, on page 7, the authors state that “we were unable to purify full PTPRU-ICD (D1 +D2)” and they were unable to generate chimeras containing PTPRU D2.

> It is true we were unable to express and purify soluble PTPRU using the same domain boundaries and conditions as for close paralogs PTPRK and PTPRM D1+D2 domains. We were able to express and purify a chimera between PTPRK-D1+PTPRU-D2, but not the reciprocal PTPRU-D1+PTPRK-D2 chimera. We have now included mammalian expression experiments to overcome this limitation, as outlined below.

Most of their analysis of potential activity in PTPRU was conducted on a truncated construct expressing the isolated D1 domain as a recombinant protein in E. coli. How do they know that the protein folded properly?

> Most other PTPs have successfully been studied as an isolated domain. However, we do take the reviewer’s point on board as detailed further below. However, as mentioned above, melt curves and limited proteolysis both support that the PTPRU-D1 domain adopts a folded conformation in solution. In addition, our PTPRU X-ray crystal structure overlays with all previous PTP domain structures with RMSDs of 0.9-1.2 Å, indicating a high degree of alignment (Supplementary Table 2). Together, these data indicate that the domain studied is a folded protein.

There have been reports to suggest a chaperone function for D2 domains – perhaps the presence of D2 is essential for manifestation of catalytic activity in PTPRU?

> This is a valid concern. Previous studies have mostly indicated an inhibitory effect of RPTP D2 domains on D1 domains (see most recently Wen et al. 2020). However, to test the impact of the PTPRU D2 domain on activity, we expressed and immunoprecipitated PTPRK and PTPRU intracellular domains (New Fig. 1f), along with corresponding cysteine mutants, from HEK293T cells. These full ICD domains were then used in pNPP activity assays and the data show that wildtype, but not mutant, PTPRK is catalytically active. In contrast,

PTPRU wildtype and mutant full ICDs displayed no detectable activity above control, even after 24 hours, and despite higher levels of protein than PTPRK (Fig 1g-h).

Also, it was reported by Philip Cohen's lab that, at least for protein phosphatase 1, the activity of the protein expressed in *E. coli* possessed different features from the activity of the phosphatase purified from mammalian tissues [MacKintosh et al (1996) FEBS Letts 397: 235-238]. With this in mind, the presence of disordered segments in the structure of the recombinant protein may reflect incorrect folding in *E. coli*.

> As indicated in the previous point we have now tested both bacterial and mammalian expressed proteins, including constructs encompassing the entire ICD, in activity assays. The lack of phosphatase activity for all PTPRU samples, in particular those from mammalian cells, suggests that expression of PTPRU in bacterial cells did not render it inactive. Therefore, although we do not have a crystal structure of this domain expressed in mammalian cells, and so cannot be certain that post-translational modifications or equivalent may influence the fold, the lack of detectable phosphatase activity is not a product of the expression system.

Reviewer #3 (Remarks to the Author):

In the manuscript, "The receptor PTPRU is a redox sensitive pseudophosphatase," Hay et al provide evidence that the protein tyrosine phosphatase D1 domain (PTP D1) of the receptor phosphatase PTPRU lacks catalytic activity. The authors also show that the protein's D1 domain is less thermostable than a homologous PTP domain and they solve the first crystal structure of PTPRU's PTP D1 domain, uncovering deviations from canonical PTP-domain structure that are consistent with PTPRU D1's lack of activity and instability. Finally, the authors provide preliminary evidence that that the intracellular portion of PTPRU can associate with substrates of other receptor PTPs (e.g., p120-catenin), sequestering them from potential dephosphorylation; when PTPRU is knocked out, the tyrosine phosphorylation of p120-catenin counterintuitively decreases.

Receptor PTPs are key regulators of cell signaling, and the demonstration that a member of the family is not really a PTP (i.e., a pseudophosphatase) would represent a significant finding. The manuscript is well written and the experiments are well designed and clearly presented. However, a revised version of the manuscript would need to address the following critiques before I could recommend it for publication.

> Thank you for this clear summary of our work.

Critique 1: Cropping of anti-phosphotyrosine westerns (Figures 1E, 1G, and 5E). The anti-pTyr blots purport to demonstrate the lack of activity of the PTPRU D1 against a whole range of potential substrates. So, why are the blots cropped at seemingly arbitrary molecular weight cut-offs? The authors should present the uncropped blots to show the tyrosine-phosphorylation levels over the entire molecular-weight range of proteins in the treated lysates.

> We initially cropped these blots to ensure we could probe for all appropriate controls including α -tubulin. To ensure full transparency we have performed repeat experiments and identified a sequence of antibody incubations that have enabled us to both detect pTyr and all controls on the same uncut membrane. These data are now presented in figures 1e, 1j and 3b.

Critique 2: How have the authors controlled for the possibility that the lack of activity in their PTPRU D1 constructs may be an artifact of truncating the protein? The thermal instability of

the protein—as well as the mysterious inability to express the D1+D2 construct—suggest that this protein doesn't “like” to be truncated. The authors should attempt experiments in which they assess activity in the context of the full-length protein. Perhaps immunoprecipitating PTPRU side-by-side with another RPTP and then comparing the PTP activities in the IPs?

> This is an excellent suggestion. We have now overexpressed flag-tagged PTPRK and PTPRU, wildtype and cysteine mutants in the context of the full intracellular domain (new Fig 1f) in HEK293T cells. These were immunoprecipitated and incubated with pNPP to monitor phosphatase activity. In this expression system, PTPRU appears to be more stable/better expressed than PTPRK (Fig 1g). But, despite more PTPRU being present in these assays, phosphatase activity was only detected for wildtype PTPRK (Fig 1h and Supplementary Fig. 3). This suggests that the lack of activity of the PTPRU D1 domain is not due to the truncated construct used, as the full ICD including the juxtamembrane and D2 domains of PTPRU still displays no activity.

Critique 3: The authors should more fully engage the previous literature that suggests that PTPRU is an active phosphatase. The authors state only in passing in the introduction that “... previous work suggests that PTPRU can dephosphorylate substrates such as β -catenin.” Indeed, one of the papers that the manuscript references (Yan et al, 2006) shows rather convincing evidence that expression of PTPRU causes a dramatic decrease in phosphorylation of β -catenin's Y102, and that this decrease is dependent on the presence of catalytically active PTPRU (the dephosphorylation is completely lost when an inactive Cys to Ser mutant is expressed instead of the wild-type protein). How do the authors square this previous finding with the lack of PTP activity in their constructs?

Have they attempted to look at PTPRU's activity with this previously reported substrate?

> In Figure 7 of Yan et al, (JBC 2006), PTPRU and a Cys>Ser mutant were overexpressed in (probably) SW480 cells. Cells were stimulated with EGF, then β -catenin was immunoprecipitated, subjected to SDS PAGE and immunoblotted using Y-102, which is a phosphotyrosine monoclonal antibody (<https://www.cellsignal.com/products/primary-antibodies/phospho-tyrosine-mouse-mab-p-tyr-102/9416>), rather than a β -catenin site-specific antibody. These data indicate that overexpression of PTPRU leads to a complete loss of β -catenin tyrosine phosphorylation. One caveat is that the levels of expression of PTPRU WT and Cys>Ser are not reported, meaning this could be a direct effect of protein overexpression. Moreover, in Figure 4 of Liu et al. (Int J Clin Exp Pathol 2014), PTPRU knockdown in AGS and SGC7901 cells also led to decreased levels of β -catenin phosphorylation (using the same Y-102 phosphotyrosine antibody). Thus, opposite effects identified in these two papers are both attributed to PTPRU emphasizing the context dependence of phosphorylation effects upon knockdown or overexpression.

These reports are reconcilable with our data as they do not measure direct dephosphorylation by PTPRU and therefore may be identifying downstream effects attributable to PTPRU protein levels. In support of this interpretation of the available data, two interactome studies of PTPRU did not detect β -catenin as an interactor. We also previously found that PTPRK, the closest paralog to PTPRU, does not interact with β -catenin (Fearnley et al 2019). For these reasons and given the clear context dependence of phosphorylation events, we have not attempted to recapitulate these assays. We have however modified the introduction to better reflect the literature describing PTPRU modulation of β -catenin phosphorylation.

REVIEWERS' COMMENTS:

Reviewer #1 (Remarks to the Author):

The authors have done a largely admirable job at addressing the concerns of the reviewers and especially, through editing changes and changes to the figures, made the results considerably more clear.

I have only a final concern, which arises, in part, to the significantly increased clarity of the message in the current manuscript. The authors argue that the pTyr binding pocket structure is 'disrupted' and, particularly, that "This new loop orientation blocks the catalytic cysteine and would directly interfere with pTyr binding (Fig. 2e)." and "Our structural data identify multiple features that would disrupt both substrate binding and catalytic activity."

Yet, the conclusion of the manuscript is that the function of the protein is to bind and titrate phosphorylated substrates from their endogenous phosphatases: "This ability to bind substrates that overlap with active phosphatases, combined with the lack of phosphatase activity of PTPRU suggests that the likely role of PTPRU in cells is to act as a decoy receptor that sequesters substrates protecting them from dephosphorylation".

How are these apparently opposing messages reconciled?

Reviewer #3 (Remarks to the Author):

The authors have satisfactorily responded to my critiques of the original manuscript, and I am happy to recommend the revised manuscript for publication.

Response to reviewers (NCOMMS-19-37816A):

Reviewer #1 (Remarks to the Author):

The authors have done a largely admirable job at addressing the concerns of the reviewers and especially, through editing changes and changes to the figures, made the results considerably more clear.

>We thank the reviewer for acknowledging the improvements made to the manuscript in response to their constructive comments.

I have only a final concern, which arises, in part, to the significantly increased clarity of the message in the current manuscript. The authors argue that the pTyr binding pocket structure is 'disrupted' and, particularly, that "This new loop orientation blocks the catalytic cysteine and would directly interfere with pTyr binding (Fig. 2e)." and "Our structural data identify multiple features that would disrupt both substrate binding and catalytic activity."

Yet, the conclusion of the manuscript is that the function of the protein is to bind and titrate phosphorylated substrates from their endogenous phosphatases: "This ability to bind substrates that overlap with active phosphatases, combined with the lack of phosphatase activity of PTPRU suggests that the likely role of PTPRU in cells is to act as a decoy receptor that sequesters substrates protecting them from dephosphorylation".

How are these apparently opposing messages reconciled?

>This is an important point and we thank the reviewer for giving us the opportunity to clarify. We have now amended our discussion to make a clear distinction between the binding of phosphotyrosine (the catalytic substrate) and the specific proteins that are bound and dephosphorylated by active RPTPs (biological substrates). In addition, we have added a clarifying sentence highlighting our previous finding that binding of protein substrates to a PTPRK (PTPRU paralog) cysteine "trapping mutant" or its D2 domain is phosphorylation-independent (Fearnley et al., 2019). These data support that protein substrate binding can occur in the absence of phosphorylation and suggest that binding of biological substrates involves more than just insertion of the pTyr into the active site. This might explain why active PTPs are quite promiscuous at the peptide level, but show selectivity at the protein level. Thus, the inability of PTPRU to bind phosphotyrosine does not preclude its ability to bind and sequester protein substrates. Although we acknowledge that the term "substrate" is less appropriate for an inactive enzyme we felt it was even more confusing to use different terms when describing the overlapping binding partners for PTPRK and PTPRU.

Reviewer #3 (Remarks to the Author):

The authors have satisfactorily responded to my critiques of the original manuscript, and I am happy to recommend the revised manuscript for publication.

>We are pleased with the reviewer's recommendation.